# Toward an optimal contraception dosing strategy

**Brenda Lyn A. Gavina**[1,2], **Aurelio A. de los Reyes V**[1,3]*, **Mette S. Olufsen**[4],
**Suzanne Lenhart**[5], **Johnny T. Ottesen**[6,7]*

**1** Institute of Mathematics, University of the Philippines Diliman, Quezon City, Philippines, **2** Maritime Academy of Asia and the Pacific, Bataan, Philippines, **3** Biomedical Mathematics Group, Pioneer Research Center for Mathematical and Computational Sciences, Institute for Basic Science, Daejeon, Republic of Korea, **4** Department of Mathematics, North Carolina State University, Raleigh, North Carolina, United States of America, **5** Department of Mathematics, University of Tennessee, Knoxville, Tennessee, United States of America, **6** Department of Sciences, Roskilde University, Roskilde, Denmark, **7** Center for Mathematical Modeling—Human Health and Diseases, Roskilde University, Roskilde, Denmark

* adlreyes@math.upd.edu.ph (AADLRV); johnny@ruc.dk (JTO)

**Data Availability Statement:** This manuscript uses data that are extracted from Fig 1 in Welt CK, McNicholl DJ, Taylor AE, Hall JE. Female reproductive aging is marked by decreased secretion of dimeric inhibin. J Clin Endocrinol

## Abstract

Anovulation refers to a menstrual cycle characterized by the absence of ovulation. Exogenous hormones such as synthetic progesterone and estrogen have been used to attain this state to achieve contraception. However, large doses are associated with adverse effects such as increased risk for thrombosis and myocardial infarction. This study utilizes optimal control theory on a modified menstrual cycle model to determine the minimum total exogenous estrogen/progesterone dose, and timing of administration to induce anovulation. The mathematical model correctly predicts the mean daily levels of pituitary hormones $LH$ and $FSH$, and ovarian hormones $E_2$, $P_4$, and $Inh$ throughout a normal menstrual cycle and reflects the reduction in these hormone levels caused by exogenous estrogen and/or progesterone. Results show that it is possible to reduce the total dose by 92% in estrogen monotherapy, 43% in progesterone monotherapy, and that it is most effective to deliver the estrogen contraceptive in the mid follicular phase. Finally, we show that by combining estrogen and progesterone the dose can be lowered even more. These results may give clinicians insights into optimal formulations and schedule of therapy that can suppress ovulation.

## Author summary

Hormonal contraceptives composed of exogenous estrogen and/or progesterone are commonly administered artificial means of birth control. Despite many benefits, adverse side effects associated with high doses such as thrombosis and myocardial infarction, cause hesitation to usage. Our study presents an improved mathematical model for hormonal control of the menstrual cycle and applies optimal control theory to minimize total exogenous estrogen and/or progesterone dose, and determine timing of administration that lead to contraception. We observe a reduction in dosage of about 92% in estrogen monotherapy and 43% in progesterone monotherapy. Our simulations show that it is most

Metab. 1999;84:105-111, using the software Digitizelt version 2.5 by Bormann I available from https://www.digitizeit.xyz/. The extracted data has been included in the supplementary material. Optimization and parameter estimation codes are accessible through the link https://github.com/3r3nd/menstrual-cycle-project.

**Funding:** BLAG was supported by University of the Philippines Office of International Linkages, a Continuous Operational and Outcomes-based Partnership for Excellence in Research and Academic Training Enhancement (UP-OIL-COOPERATE) grant, and a Commission on Higher Education Faculty Development Program - II (CHED-FDP-II) scholarship. AADLRV acknowledges the support of the Institute of Mathematics, University of the Philippines Diliman and the Institute for Basic Science (IBS-R029-C3). The funders had no role in study design, data collection and analysis, decision to publish, or preparation of the manuscript.

**Competing interests:** The authors have declared that no competing interests exist.

effective to deliver the estrogen contraceptive in the mid follicular phase. In addition, we illustrate that combination therapy significantly lower doses further. Our findings may give clinicians insights into optimal dosing scheme for contraception.

## Introduction

The female's reproductive life spanning approximately 39 years from age of 12.5 until 51 is governed by the menstrual cycle [1], a cyclic process regulated by the endocrine system. A normal menstrual cycle involves ovarian follicular development, ovulation, and luteinization influenced by the hormones gonadotropin-releasing hormone (GnRH), luteinizing hormone ($LH$), follicle-stimulating hormone ($FSH$), estradiol ($E_2$), progesterone ($P_4$), and inhibin ($Inh$), which are produced in the hypothalamus, pituitary, and ovaries [2, 3]. During this cycle, the pituitary and ovarian hormones fluctuate. Unusual concentrations of these hormones lead to abnormal cycles. For instance, low levels of $LH$, $FSH$, and $E_2$ cause anovulation [3]. Anovulation is an abnormal menstrual cycle characterized by the absence of the ovulation process [4].

Numerous previous modeling studies have examined the menstrual cycle, how it is formed and how it can be altered. The most significant body of work stems from Selgrade et al. [5–13]. These studies start by developing a continuous menstrual cycle model and fitting it to data. This model consists of two parts tracking the pituitary [6] and ovarian [5] portions of the cycle. The full model developed by Harris et al. [7, 8, 12] is formulated as an autonomous nonlinear system of 13 delay differential equations (DDEs) merging the pituitary and ovarian components. The model, consisting of positive and negative feedback relationships of the pituitary and ovarian hormones, was fitted to data from McLachlan et al. [14] reporting daily blood concentrations of $LH$, $FSH$, $E_2$, $P_4$, and $Inh$ averaged from data for 33 normally cycling women. Pasteur [9] expanded this model distinguishing the effect of inhibin A ($Inh\ A$) and inhibin B ($Inh\ B$). The inclusion of the two forms of inhibin provides a more realistic representation of the human menstrual cycle since these hormones are active in different phases of the cycle [9]. The model was further developed in the study by Margolskee et al. [11] who reduced the number of delays to one (delay in inhibin) and fitted it to data by Welt et al. [15], which consist of daily mean blood levels of $LH$, $FSH$, $E_2$, $P_4$, and $Inh\ A$ averaged over 23 normally cycling women. Most recently, Wright et al. [13] added autocrine mechanisms to the Margolskee model to describe the influence of exogenous estrogen and/or progesterone in perturbing a normal cycle to contraceptive state.

In addition to work by Selgrade and collaborators, Chen et al. [16] introduced a simple model with three delay differential equations describing the hormonal interactions of the human menstrual cycle along the hypothalamus-pituitary-ovaries axis. Reinecke et al. [17, 18] studied the pulsatile release of GnRH in the hypothalamus using a stochastic process, and Röblitz et al. [19] utilized ordinary differential equations (ODEs) to examine the interactions between GnRH, pituitary, and ovarian hormones which shortened computational time.

These menstrual cycle models and their modifications, all fitted to biological data consisting of blood levels of pituitary and ovarian hormones from normally cycling women, have been used to study fertility [7, 9, 12, 19], reproductive diseases [16], and contraception [13, 17–19]. For instance, Pasteur's model showed that the dosage of exogenous estrogen varies inversely as the amplitudes of hormone level fluctuations, while Chen's model simulated treatment of uterine myomas with GnRH analogues.

Contraception is achieved through natural and/or artificial means. Artificial methods include hormonal, barrier, permanent, and long-acting reversible contraceptives. Today,

contraception is most commonly achieved by taking a daily pill, though this method is rapidly being replaced by injectables and implants [20]. Independent of the administration method, almost all hormonal contraceptives including exogenous progesterone and/or estrogen act by blocking ovulation, changing cervical mucus, which hinders sperm transport and/or modifying endometrium which prevents implantation [21]. Aside from contraceptive benefit, suppression of ovulation can alleviate negative premenstrual symptoms [22–24] and reduce anterior cruciate ligament (ACL) injury [25, 26], among others. For example, Hammarbäck et al. [23] showed that cyclical negative premenstrual symptoms such as irritability and breast tenderness disappear in anovulatory cycles. In addition, Yonkers et al. [24] found that administration of GnRH analogues, exogenous estrogen, and certain oral contraceptives at doses which inhibit ovulation is effective in reducing the symptoms. The paper [27] reported that ACL injuries in female athletes are significantly greater during the ovulatory phase and studies [25, 28, 29] suggested that oral contraceptive users have an almost 20% decreased risk for ACL injury. Most literature studies [13, 18, 19] focus on the administration of exogenous hormones such as estrogen, progesterone, and GnRH analogues to inhibit ovulation. For example, Reinecke's detailed DDE model included a numerical study on the parts of body affected by estrogen and/or progesterone hormonal contraceptives. This model compared results from administering the contraceptive continuously at a constant rate and at certain time points. Röblitz's ODE model compares the effect of administering a single and multiple dose of Nafarelin (a GnRH agonist) and Cetrorelix (a GnRH antagonist known to impede ovulation in invitro fertilization treatment), and Wright's model simulated exogenous estrogen and/or progesterone doses concluding that by combining the two effectors the dose can be lowered significantly. These studies explored the effect of exogenous hormones for inducing anovulation but did not examine optimal dosing. With rapid advance in implants and injections providing continuous administration there is great potential to implement new patient-specific minimizing dosing schemes. To our knowledge, our work is the first to use modeling to study timing of dosing thereby minimizing the dose even more. As implants become more common, results from this study have the potential to provide contraception to more women, in particular since lower doses also decrease the risks for adverse side effects such as venous thromboembolism and myocardial infarction associated with high doses of hormonal contraceptives [3, 30–32].

While optimal control theory has not been used to simulate contraception, the theory has long tradition in biology to find strategies that optimize an outcome. In [33], the theory is applied to a system of ODEs to determine a scheme for delivery of insulin and glucagon that regulates blood glucose level in diabetes patients. The study [34] utilized optimal control to develop treatment protocol for tumor stabilization for prostate cancer. Several studies explore optimization of hormonal treatments. For instance, [35] described an optimal dosing regimen for the infusion of *FSH* to patients undergoing in vitro fertilization. While in [36], control theory is employed to investigate optimal dosage decisions in the administration of gonadotropin in controlled ovarian hyperstimulation treatment cycle. The current paper expands previously published papers by Selgrade et al. on the hormonal regulation of the menstrual cycle [7, 12] and the transition to contraception [13]. In this study, the model in Margolskee and Selgrade [11] is modified to include mechanisms depicting the contraceptive effect of exogenous progesterone on the menstrual cycle. This new model shows the principal mechanisms behind transition to contraception. It is calibrated to the patient-data extracted from Welt et al. [15] and predicts the daily levels of pituitary hormones *LH* and *FSH*, and ovarian hormones $E_2$, $P_4$, and *Inh* averaged during a normal menstrual cycle in 23 women. The model output also predicts reduction in pituitary and ovarian hormone levels caused by exogenous estrogen and/or progesterone observed by Obruca et al. [37] and Deb et al. [38].

This paper uses an optimal control approach to simulate contraception using the model described above. The objective is to identify strategies to understand when and how much estrogen and/or progesterone to administer to obtain a contraceptive state. Results show that the dosage may be reduced by 92% in estrogen monotherapy, 43% in progesterone monotherapy, to suppress ovulation. Simulations agree with biological literature that in monotherapy, administration of estrogen in the mid follicular phase is effective at preventing ovulation. Lastly, numerical experiments show that by combining estrogen and progesterone the dose can be reduced even further. The results of this study may aid in identifying the minimum dose and treatment schedule that cause anovulation.

## Materials and methods

This section describes the normal and anovulatory menstrual cycle, data, the mathematical model, parameter estimation, and optimal control method. Table 1 lists the state variables in the mathematical model. The model parameters and their values are given in Table B in S1 Text.

### The normal menstrual cycle

The normal menstrual cycle (shown in Fig 1) for an adult female has an average length of 28 days. It has two stages, the follicular phase and luteal phase. Through hormones, the hypothalamus, pituitary, and ovaries interact to regulate the menstrual cycle [2, 3].

During the menstrual cycle, the hypothalamus produces pulses of GnRH which control pituitary's secretion of the gonadotropins $FSH$ and $LH$. At the beginning of the follicular phase (the start of menstruation or menses), $FSH$ rises and causes the recruitment of a group of immature follicles. As these follicles develop through the stimulation of the gonadotropins $FSH$ and $LH$, they increase secretion of $E_2$ (see Fig 2). Follicular development is indicated by an enlargement of oocyte (immature egg), multiplication, and transformation of granulosa cells (structure that surrounds the oocyte) to a cuboidal shape, and formation of small gap junctions, which enable nutritional, metabolite, and signal interchange between the granulosa cells and oocyte [3]. Toward the end of the follicular stage, one dominant follicle continues to grow while the rest of the follicles become atretic. As the dominant follicle grows, the $E_2$ level

**Table 1. Menstrual cycle state variables and initial conditions.**

| State variable | Description | Initial condition | Unit | Reference |
|---|---|---|---|---|
| $RP_{LH}$ | Amount of reserve pool $LH$ | 167.57 | IU | estimated |
| $LH$ | $LH$ blood concentration | 11.81 | IU/L | estimated |
| $RP_{FSH}$ | Amount of reserve pool $FSH$ | 14.48 | IU | estimated |
| $FSH$ | $FSH$ blood concentration | 11.41 | IU/L | estimated |
| $RcF$ | Amount of active follicular mass in the recruited follicular stage | 2.10 | ng | estimated |
| $GrF$ | in the growing follicular stage | 4.12 | ng | estimated |
| $DomF$ | in the dominant follicular stage | 0.46 | ng | estimated |
| $Sc_1$ | during ovulation | 1.06 | ng | estimated |
| $Sc_2$ | during luteinization | 1.67 | ng | estimated |
| $Lut_1$ | Amount of active luteal mass in the first luteal stage | 4.16 | ng | estimated |
| $Lut_2$ | in the second luteal stage | 13.03 | ng | estimated |
| $Lut_3$ | in the third luteal stage | 16.48 | ng | estimated |
| $Lut_4$ | in the fourth luteal stage | 10.29 | ng | estimated |

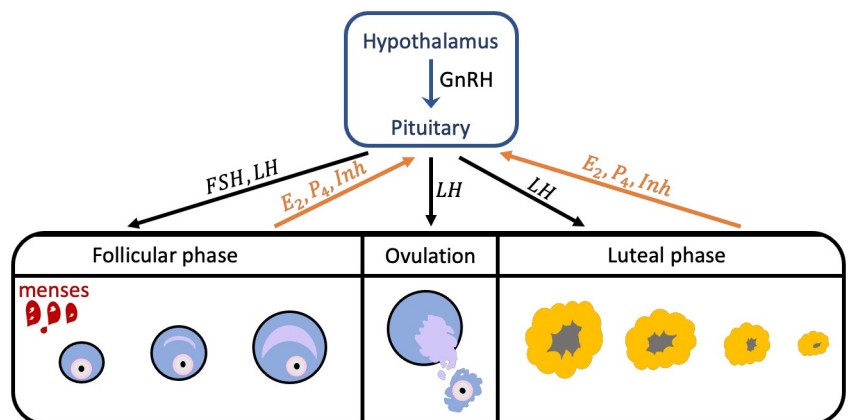

**Fig 1. The Follicular and Luteal Phases of the menstrual cycle.** The figure shows the transition of a follicle, from its growth in the follicular phase to its rupture during ovulation, as well as its transformation to a corpus luteum and degradation in the luteal phase. The blue arrow indicates the control of the hypothalamus in the secretion of pituitary hormones. The black arrows represent the influence of the pituitary system on the ovarian system through $FSH$ and $LH$, and the orange arrows show the response of the ovarian system through $E_2$, $P_4$, and $Inh$.

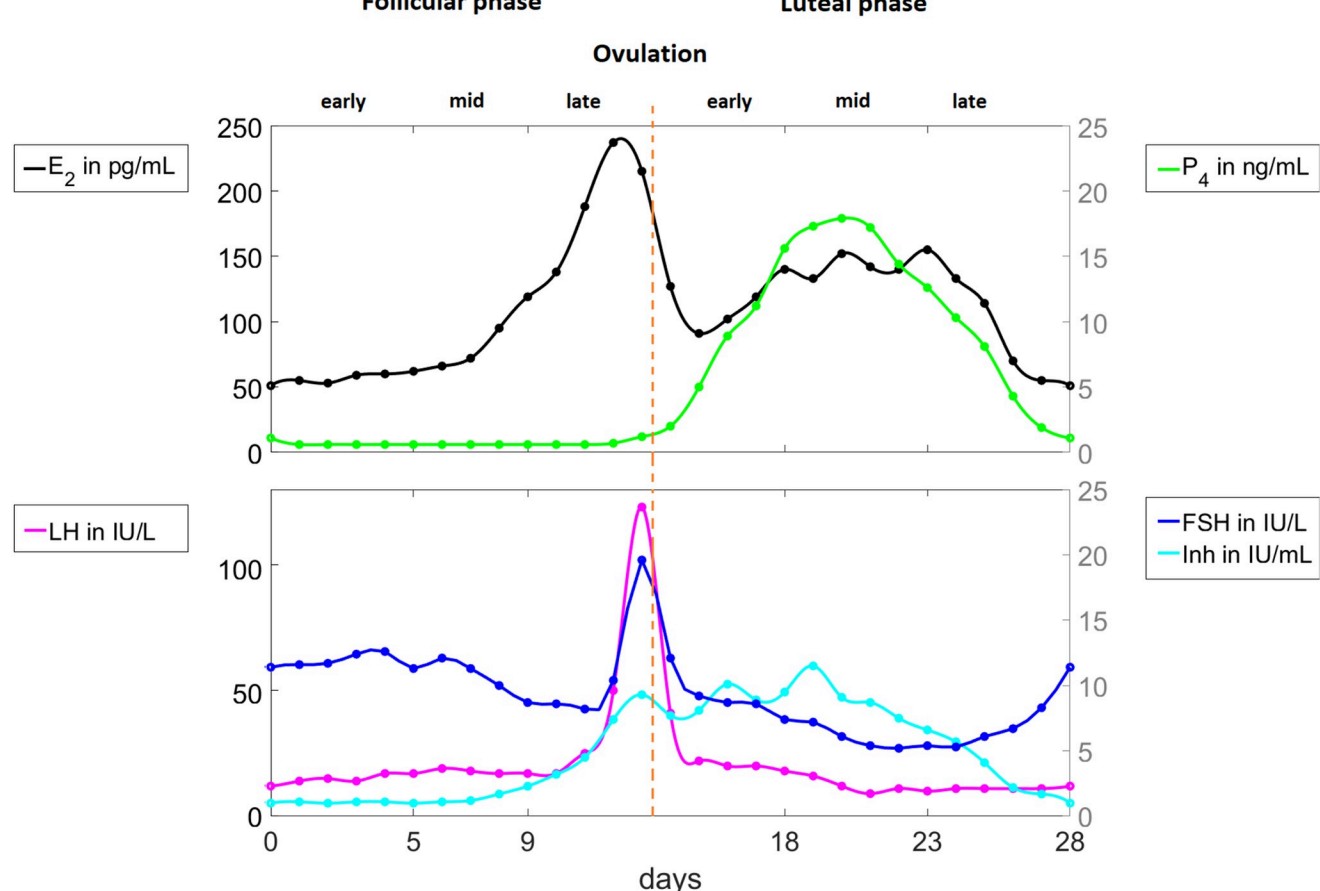

**Fig 2. Pituitary and ovarian hormone levels in a normal menstrual cycle.** Data digitized from the study by Welt et al. [15] is interpolated by cubic splines. The hormones $LH$, $FSH$, and $E_2$ peak in the late follicular phase while $P_4$ and $Inh$ reach maximum value in the luteal phase.

increases to a maximum value prompting an *LH* surge. The *LH* surge stimulates ovulation, releasing the egg from the dominant follicle. The ruptured dominant follicle then transforms to a corpus luteum. As the corpus luteum grows, $P_4$ and *Inh* production is increased. $P_4$ and *Inh* inhibit the synthesis of *LH* and *FSH*. If fertilization does not occur, the corpus luteum degrades removing the inhibition on *LH* and *FSH*. Consequently, levels of *FSH* and *LH* rise and the menstrual cycle repeats [2, 3].

## The anovulatory cycle and contraception

Anovulation occurs when the follicle fails to release the egg. This state can be detected by measuring serum progesterone. During a normal menstrual cycle, progesterone is largely produced by the corpus luteum after ovulation. Its levels stay below 2 ng/mL during the follicular phase and peak 7 to 8 days after ovulation [3]. Accordingly, a low concentration of this hormone in the luteal phase indicates anovulation or defective luteinization [39]. The menstrual cycle is classified as anovulatory if progesterone concentration remains below 5 ng/mL without an *LH* peak [40]. In fact, disruption at any level of the hypothalamic-pituitary-gonadal axis can result in anovulation. This includes suppression of GnRH, the presence of a pituitary tumor leading to gonadotropin secretory dynamics that fail to stimulate follicular growth, or abnormal estrogen feedback signaling causing inhibition of *FSH* secretion or a low estrogen level which prevent the *LH* surge [3].

Hormonal contraceptives, composed of progesterone alone, or a combination of estrogen and progesterone are widely used artificial means of contraception. They are delivered orally, transdermally, vaginally, via implants, or injections. Estrogen or progesterone alone can cause contraception via anovulation but the combined administration of both hormones significantly enhances effectiveness [41].

In oral steroid contraceptives composed of both exogenous estrogen and progesterone, progesterone induces anovulation by preventing the midcycle rise in *LH* secretion [41, 42]. This is due to progesterone's suppression of follicular development and gonadotropin secretion [43]. The insufficient *FSH* prevents follicle growth. Consequently, the lack of follicle growth results in an inadequate amount of estradiol inhibiting the *LH* surge [3]. The estrogen component suppresses *FSH* secretion blocking folliculogenesis, stabilizing the endometrium, which minimize bleeding [42, 44].

## Data

The output of the mathematical model without the administration of exogenous hormone is compared to the 28-day data (solid circles in Fig 2) from Welt et al. [15]. These data (see Table A in S1 Text), extracted from Fig 1 in [15] using the software DigitizeIt version 2.5 [45], comprise mean levels of $E_2$, $P_4$, *Inh A*, *LH*, and *FSH* taken from 23 normally cycling younger women aged 20 to 34 years, all with a history of a regular 25–35 day menstrual cycle with evidence of ovulation in the preceding cycle. Our study employs only *Inh A* since *Inh B* is more significant in studies about reproductive aging. The Welt study [15] reports that the data are centered to the day of ovulation requiring three of the following four criteria to be satisfied: i) LH peak day, ii) midcycle FSH peak day, iii) the day of or after the midcycle $E_2$ peak, and iv) day that the $P_4$ doubled from its baseline or reached a level of 0.6 ng/mL. Before averaging the data the menstrual cycle data were standardized to a 28-day cycle length with the day of ovulation centered to day 0 and the mean hormone levels were averaged over the early, mid-, and late follicular phase and early, mid-, and late luteal phase, see [15] for more details.

## Mathematical model of the normal menstrual cycle

To study anovulation, we use the normal menstrual cycle model by Margolskee et al. [11] and induce anovulation via added estrogen and progesterone. The core model, shown in Fig 3, includes the pituitary and ovarian phases. It assumes that

**a.1** $LH$ and $FSH$ synthesis occurs in the pituitary,

**a.2** $LH$ and $FSH$ are held in a reserve pool awaiting release into the bloodstream, and

**a.3** the follicular/luteal mass undergoes nine ovarian stages of development.

In the model, $RP_{LH}(t)$ and $RP_{FSH}(t)$ denote the amount of $LH$ and $FSH$ in the reserve pool at time $t$ days, $LH(t)$ and $FSH(t)$ are the blood concentrations of $LH$ and $FSH$; $E_2(t)$, $P_4(t)$, and $Inh(t)$ denote the blood levels of $E_2$, $P_4$, and $Inh$; and $RcF(t)$, $GrF(t)$, $DomF(t)$, $Sc_1(t)$, $Sc_2(t)$, $Lut_1(t)$, $Lut_2(t)$, $Lut_3(t)$, and $Lut_4(t)$ are the masses of active follicular/luteal tissues in the nine ovarian stages: recruited, growing, and dominant follicular stages, ovulation, luteinization, first, second, third, and fourth stages of luteal development.

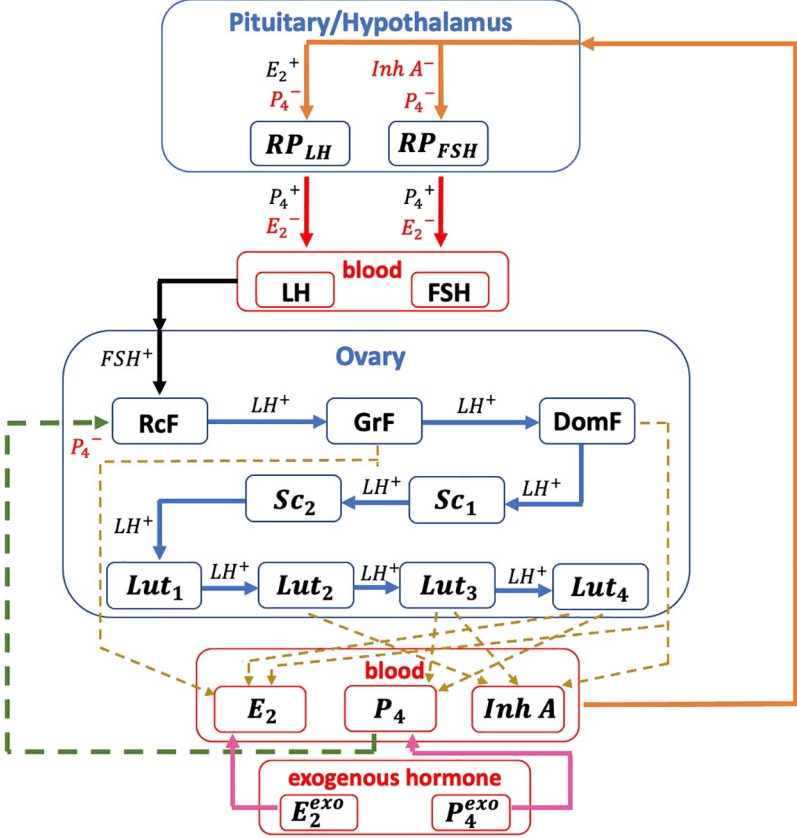

**Fig 3. Model diagram.** The diagram (adapted from [13]) shows the 13 states ($RP_{LH}$, $RP_{FSH}$, $LH$, $FSH$, $RcF$, $GrF$, $DomF$, $Sc_1$, $Sc_2$, $Lut_1$, $Lut_2$, $Lut_3$, $Lut_4$) in the menstrual cycle model. $H^+$ denotes stimulation by hormone $H$ while $H^-$ denotes inhibition. The red arrow presents the release of the pituitary hormone from the reserve pool to the blood. The black arrow indicates the influence of the pituitary system on the ovarian system while the orange arrow denotes the ovary's feedback. The green dashed arrow denotes inhibition of $P_4$ in the $RcF$ stage. The blue arrow describes the transition of a follicle from one ovarian stage to the next. The gold dashed arrow denotes the state contributing to the ovarian hormone production. The pink arrow represents the infusion of exogenous hormone to the ovarian system.

**The pituitary model** includes the hypothalamus and the pituitary. It predicts the synthesis, release, and clearance of $LH$ and $FSH$, and the pituitary's response to $E_2$, $P_4$, and $Inh$ (see Fig 3). The direct effect of the ovarian hormones on the pituitary and indirect influence via the hypothalamus are grouped partly because of the complexity of tracking GnRH [6].

The rate of change of $RP_{LH}(t)$ in Eq (1) is the difference between two terms. The first term describes the synthesis in the pituitary and the second term represents the release of $LH$ into the blood. A Hill function is used in the synthesis term of $LH$ to predict the strong stimulatory effect for $E_2$ levels above the threshold level $Km_{LH}$. $P_4$ inhibits the production but bolsters the $LH$ release into the bloodstream. $E_2$ inhibits the release of $LH$. The change in $LH(t)$ in Eq (2) is affected by the $LH$ release into and clearance from the blood. The $LH$ release rate is assumed proportional to its amount in the reserve pool, while the clearance rate is proportional to the $LH$ blood level.

Likewise, the rate of change of $RP_{FSH}(t)$ in Eq (3) is governed by synthesis and release. The synthesis of $FSH$ is inhibited by both $Inh$ and $P_4$. The time delay $\tau$ is introduced to account for the time it takes the $FSH$ synthesis rate to respond to changes in the $Inh$ concentration. $P_4$ stimulates the $FSH$ release into the bloodstream. The quadratic expression $E_2^2$ in the $FSH$ release term is included to ensure greater inhibitory effect of $E_2$ on $FSH$ release than on $LH$ release. Similarly, the rate of change of $FSH(t)$ in Eq (4) depends on two terms: a release term assumed proportional to the amount of $FSH$ in the reserve pool and a clearance term assumed proportional to the $FSH$ blood concentration.

Applying this model to the hormone cascade described above, gives the following system of DDEs

$$\frac{d}{dt}RP_{LH}(t) = \frac{V_{0,LH} + \frac{V_{1,LH}E_2(t)^8}{Km_{LH}^8 + E_2(t)^8}}{1 + P_4(t)/Ki_{LH,P}} - \frac{k_{LH}[1 + c_{LH,P}P_4(t)]RP_{LH}(t)}{1 + c_{LH,E}E_2(t)}, \tag{1}$$

$$\frac{d}{dt}LH(t) = \frac{1}{v}\frac{k_{LH}[1 + c_{LH,P}P_4(t)]RP_{LH}(t)}{1 + c_{LH,E}E_2(t)} - \alpha_{LH}LH(t), \tag{2}$$

$$\frac{d}{dt}RP_{FSH}(t) = \frac{V_{FSH}}{1 + Inh(t-\tau)/Ki_{FSH,Inh} + P_4(t)/w} - \frac{k_{FSH}[1 + c_{FSH,P}P_4(t)]RP_{FSH}(t)}{1 + c_{FSH,E}E_2(t)^2}, \tag{3}$$

$$\frac{d}{dt}FSH(t) = \frac{1}{v}\frac{k_{FSH}[1 + c_{FSH,P}P_4(t)]RP_{FSH}(t)}{1 + c_{FSH,E}E_2(t)^2} - \alpha_{FSH}FSH(t). \tag{4}$$

**The ovarian model** predicts the response of the ovarian hormones $E_2$, $P_4$, and $Inh$ as functions of $LH$ and $FSH$.

The rate of change of the mass of active follicular/luteal tissue in Eqs (5) to (13) depends on the mass of follicular/luteal tissue promoted to that stage and the mass advancing to the next stage. $FSH$ in Eq (5) stimulates, while $P_4$ inhibits, the recruitment of immature follicles to the $RcF$ stage. $LH$ in Eqs (5) to (7) aids the growth and transition of follicles to the next follicular stage until ovulation.

The rate of production of hormones at each ovarian stage is assumed proportional to the active mass of the follicle or corpus luteum at that stage. Furthermore, the blood concentrations of the ovarian hormones are assumed at a quasi-steady state because the clearance of ovarian hormones from the blood is faster than the clearance of pituitary hormones and the time scale for follicular and luteal development. Thus, the blood concentration of each ovarian hormone in Eqs (14) to (16) is written as linear combinations of follicular/luteal masses in the

stages secreting it [5, 11, 12, 17]. The constants $b_1$ and $b_2$ in Eqs (14) and (15) reflect the functions of exogenous estrogen $E_2^{\text{exo}}(t)$ and progesterone $P_4^{\text{exo}}(t)$, given as blood concentrations, which may be different from endogenous hormones. The exact influence of the exogenous hormones on the endogenous hormone levels are not incorporated and thus we let $b_1 = 1$ and $b_2 = 1$ agreeing with the mathematical model published in [7, 12, 13]. Eqs (5) to (16) reflect the transition of follicular/luteal mass from recruitment, to ovulation, to the stages of the luteal phase, and the effect of the pituitary to the ovarian hormones, given by the following ordinary differential equations

$$\frac{d}{dt}RcF(t) = (b + c_1 RcF(t))\frac{FSH(t)}{1 + P_4(t)/q} - c_2 LH(t)^\alpha RcF(t), \tag{5}$$

$$\frac{d}{dt}GrF(t) = c_2 LH(t)^\alpha RcF(t) - c_3 LH(t)GrF(t), \tag{6}$$

$$\frac{d}{dt}DomF(t) = c_3 LH(t)GrF(t) - c_4 LH(t)^\gamma DomF(t), \tag{7}$$

$$\frac{d}{dt}Sc_1(t) = c_4 LH(t)^\gamma DomF(t) - d_1 Sc_1(t), \tag{8}$$

$$\frac{d}{dt}Sc_2(t) = d_1 Sc_1(t) - d_2 Sc_2(t), \tag{9}$$

$$\frac{d}{dt}Lut_1(t) = d_2 Sc_2(t) - k_1 Lut_1(t), \tag{10}$$

$$\frac{d}{dt}Lut_2(t) = k_1 Lut_1(t) - k_2 Lut_2(t), \tag{11}$$

$$\frac{d}{dt}Lut_3(t) = k_2 Lut_2(t) - k_3 Lut_3(t), \tag{12}$$

$$\frac{d}{dt}Lut_4(t) = k_3 Lut_3(t) - k_4 Lut_4(t), \tag{13}$$

with auxiliary equations

$$E_2(t) = e_0 + e_1 GrF(t) + e_2 DomF(t) + e_3 Lut_4(t) + b_1 E_2^{\text{exo}}(t), \tag{14}$$

$$P_4(t) = p_0 + p_1 Lut_3(t) + p_2 Lut_4(t) + b_2 P_4^{\text{exo}}(t), \tag{15}$$

and

$$Inh(t) = h_0 + h_1 DomF(t) + h_2 Lut_2(t) + h_3 Lut_3(t). \tag{16}$$

## Novel model modifications: Anovulatory cycle and contraception

Exogenous estrogen and/or progesterone inhibit ovulation through different mechanisms. Estrogen causes anovulation by suppressing gonadotropin secretion, depicted in the release term of Eqs (1) and (3). Low gonadotropin levels inhibit maturation of follicles causing low

production of estrogen insufficient to induce an *LH* surge. The administration of progesterone reduces ovarian hormone levels [37]. The Margolskee model [11] is unable to reflect this condition. Therefore, to attain contraceptive effect of progesterone we included the following terms accounting for

**b.1** $\frac{P_4(t)}{w}$ in the synthesis term of Eq (3) and

**b.2** $\frac{P_4(t)}{q}$ in the first term of Eq (5).

The term $\frac{P_4(t)}{q}$ describes the direct inhibitory action of progesterone on follicular development [43, 46]. Baird et al. [43] and Setty et al. [46] also suggested that reduced follicle growth is further caused by the suppression of gonadotropin secretion. Moreover, Batra et al. [47] found that in ovine pituitary cell culture, progesterone decreases *FSH* secretion through decreased *FSH* biosynthesis. This inhibition of *FSH* production is accounted for by the term $\frac{P_4(t)}{w}$. In the model, the administration of both $E_2^{\text{exo}}(t)$ and $P_4^{\text{exo}}(t)$ increases suppression of gonadotropin secretion. This leads to lower combined doses to induce anovulation, showing the enhanced effectivity of combined treatment suggested in Rivera et al. [41].

## Parameter estimation

To calibrate the model to the data extracted from Welt et al. [15], we estimated selected model parameters as follows:

**c.1** The parameters in the pituitary model (Eqs (1), (2) and (3) (without $P_4(t)/w$, and (4)) are estimated starting from parameter values and initial conditions in [11]. For this submodel, we replaced $E_2(t)$, $P_4(t)$, and *Inh*(t) by time-dependent functions fitted to $E_2$, $P_4$, and *Inh* levels in the data extracted from Welt et al. [15].

**c.2** The parameters in the ovarian model (Eqs (5) (without $P_4(t)/q$) to (13), (14) (with $E_2^{\text{exo}}(t) = 0$), (15) (with $P_4^{\text{exo}}(t) = 0$), and (16)) are estimated starting with parameter values and initial conditions in [11]. For this submodel, we substituted *LH*(t) and *FSH*(t) by time-dependent functions fitted to *LH* and *FSH* levels in data extracted from Welt et al. [15].

**c.3** The parameters obtained in c.1 and c.2 are used to estimate the parameters in the merged pituitary and ovarian model.

**c.4** We do not have physiological knowledge of the parameters *w* and *q* thus, values assigned to *w* and *q* with order of magnitude between 0 and 1, and the parameter set obtained in c.3 are used as initial guess to estimate parameters in the final merged model.

In **c.1** to **c.4** we used the MATLAB `fminsearch` function, which utilizes the Nelder-Mead simplex algorithm, to estimate parameters that minimize least squares error. The least squares function employed in c.4 is

$$\frac{1}{M-N} \sum_{i=1}^{M} \left( w_{E_2^*}(i) \left( (E_2(i) - E_2^*(i)) \frac{1}{E_2^*(i)} \right)^2 + w_{P_4^*}(i) \left( (P_4(i) - P_4^*(i)) \frac{1}{P_4^*(i)} \right)^2 \right.$$

$$+ w_{Inh^*}(i) \left( (Inh(i) - Inh^*(i)) \frac{1}{Inh^*(i)} \right)^2 + w_{LH^*}(i) \left( (LH(i) - LH^*(i)) \frac{1}{LH^*(i)} \right)^2$$

$$\left. + w_{FSH^*}(i) \left( (FSH(i) - FSH^*(i)) \frac{1}{FSH^*(i)} \right)^2 \right),$$

where $H(i)$ is the hormone model output at day $i$, and $H^*(i)$ is the hormone Welt data at day $i$ for hormones $E_2$, $P_4$, $Inh$, $LH$, and $FSH$. The weight $w_{H^*}(i)$ equals 1 except for:

i. $w_{E_2^*}(13) = 1.58$, the z-score of 13th $E_2$ data (the maximum $E_2$ data),

ii. $w_{P_4^*}(21) = 1.39$, the z-score of 21th $P_4$ data (the maximum $P_4$ data),

iii. $w_{Inh^*}(20) = 1.35$, the z-score of 20th $Inh$ data (the maximum $Inh$ data),

iv. $w_{LH^*}(14) = 2.16$, the z-score of 14th $LH$ data (the maximum $LH$ data), and

v. $w_{FSH^*}(14) = 1.80$, the z-score of 14th $FSH$ data (the maximum $FSH$ data).

$M$ is the number of data points and $N$ is the combined number of model parameters and initial conditions. The Welt data was standardized to 28 days. We used four repetitions of this data as shown in Fig 4 to obtain $M$ data points utilized in parameter estimation. This is done to obtain a periodic solution. The difference $M - N$ in the denominator ascertains that errors don't increase with repetition of the data set. Because the hormones have different units, the multiplier $\frac{1}{H^*(i)}$ is used to obtain the term $(H(i) - H^*(i))\frac{1}{H^*(i)}$, the percentage error for hormone $H$. This eliminates the order of magnitude difference between the variables, allowing addition of residuals in the objective function. The *z−score*, denoting the number of standard deviations a data point is from the mean, aids in approximating the maximum hormone data. Manual adjustments after optimization are carried out to reach the maximum $E_2$, $P_4$, $Inh$, $LH$, and $FSH$, and to obtain cycle length close to the data cycle length of 28 days. This is essential since an anovulatory cycle is determined by the decrease in $P_4$ and $LH$ maximum levels from the values in a normal cycle. The resulting initial conditions and parameter values are shown in Table 1 and Table B in S1 Text.

## Optimal control applied to the menstrual cycle model

Optimal control theory describes control strategies to steer a system towards an optimal outcome specified in a cost function [48, 49]. In an optimal control problem, state variables $x(t)$ depending on time $t$, model a dynamical system and appropriate control function $u(t)$ is obtained by optimizing an objective function $J(u)$ subject to constraints [49]. In particular, consider the optimal control problem which minimizes an objective function $J(u)$ to determine optimal $u(t)$ and corresponding $x(t)$. Mathematically, it can be written as

$$\min_{u(t)} J(u) = \min_{u(t)} \int_{t_0}^{t_f} f(t, x(t), u(t))\, dt$$

subject to $x'(t) = g(t, x(t), u(t))$ and $x(t_0) = x_0$.

In this study, we let $u_1(t) = E_2^{\text{exo}}(t)$, $u_2(t) = P_4^{\text{exo}}(t)$, and

$$x(t) = (RP_{LH}(t), LH(t), RP_{FSH}(t), FSH(t), RcF(t), GrF(t),$$
$$DomF(t), Sc_1(t), Sc_2(t), Lut_1(t), Lut_2(t), Lut_3(t), Lut_4(t)).$$

The objective is to seek minimum dosage for $u_1(t)$ and $u_2(t)$ which decreases the $P_4$ peak to a value resulting in an anovulatory state. That is,

$$\min_{u_1(t), u_2(t)} J(u) = \min_{u_1(t), u_2(t)} \int_{t_0}^{t_f} \left((P_4(t) - P_0)^2 + a_1 u_1 + a_2 u_2^4\right) dt$$

subject to the model $x'(t) = g(t, t - \tau, x(t), u_1(t), u_2(t))$ and $x(t_0) = x_0$.

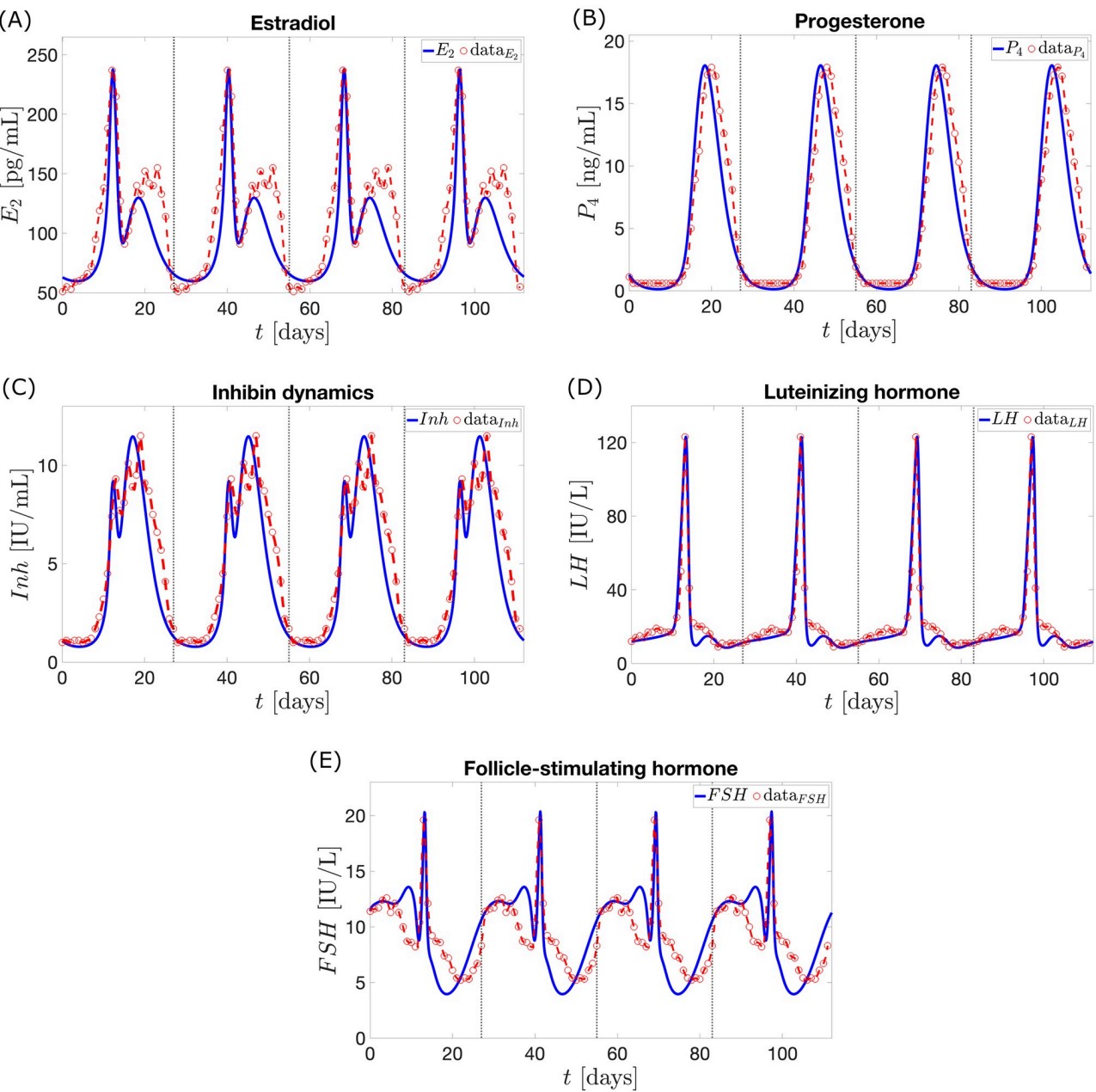

**Fig 4. Normal cycle solution.** The blue curves describe the dynamics of the pituitary and ovarian hormones predicted by the model without exogenous estrogen and progesterone. The vertical lines partition the red curves into four cycles. Each partition presents the 28-day normal cycle hormone data extracted from Welt et al. [15].

The term $(P_4(t) - P_0)^2$ in the integrand is standard for getting $P_4(t)$ to track the target $P_0$ [50]. Because we want to keep $P_4(t) < 5$ ng/mL for $0 = t_0 < t_f = 28$, a target point $P_0$ must be lower than 5 ng/mL. In addition, simulations for constant dosage show that lowering $P_4$ level further requires higher dose of exogenous hormones. Thus, we take $P_0 = 4$ ng/mL, a value lower than but close to 5 ng/mL as target point. The second and third terms are included to minimize the dosage of exogenous hormones. The power of $u_2$ is 4 in order to add convexity to the optimal control problem and to smoothen the control. We have made an exhaustive investigation

of the powers of $u_1$ and $u_2$ in the objective function which presented results with least oscillation (few examples are shown in Fig C to Fig H in S1 Text). To identify the best objective function, we started with linear $u_1$ and $u_2$ cost terms. However, this form produced oscillatory results. Several runs revealed that the oscillations where the highest for optimal progesterone results. Thus, we increased the exponent of $u_2$ in the cost function and settled for a fourth power to smoothen the optimization results. To further smoothen results other methods may be explored like varying initial conditions and number of time points. The constants $a_1 > 0$ and $a_2 > 0$ are weight factors which balance the effort of minimizing $u_1$ and $u_2$, and hitting $P_0$. Random values with order of magnitude between $-2$ and $1$ are originally assigned as initial guesses for the weights. If these weights produced $P_4$ peak higher than 5 ng/mL, their values are adjusted down. This iterative procedure was repeated until weights that make maximum $P_4(t) < 5$ ng/mL are obtained. We found that low values of weights produce small value of $\int_{t_0}^{t_f} (P_4(t) - P_0)^2 dt$ (see Table D in S1 Text), i.e., $P_4(t)$ is near $P_0$, but increases exogenous hormone dose.

To solve the optimal control problem we applied control parameterization. In this numerical method the control function is approximated by a linear combination of basis functions [48, 51]. Specifically, we utilize the MATLAB `makima` function which performs a piecewise cubic Hermite interpolation to estimate the control function. Moreover, we use `dde23` to solve the delay differential equations and `fmincon` to generate components of the piecewise cubic polynomial which gives the local least cost.

## Exploring effects of exogenous hormones

To investigate the effect of exogenous estrogen and progesterone on the menstrual cycle, we compared

**d.1** model output without exogenous hormone to data extracted from Welt et al. [15] for normally cycling women repeated four times,

**d.2** model output with constant dose of exogenous hormone to hormone output in d.1, and

**d.3** model output with optimal time-varying dose of exogenous hormone to hormone outputs in d.1 and d.2.

Some types of hormonal contraceptives like implants, injections, and patches are administered non-orally and continuously [21] while birth control pills are taken orally at specific time points. In **d.2** and **d.3** we study model response to exogenous estrogen and progesterone monotherapy, and combined treatment administered continuously for one cycle (28 days).

This study suggests a method/tool on how ovulation can be suppressed for an example woman with a specific cycle length. Though we are not presenting a population study, the method we described can be repeated to any woman if her hormone levels are known. To illustrate the adaptability of our method to women with different cycle lengths and understand how predictions change with variation in the period, we performed

**e.1** sensitivity analysis (shown in Fig A in S1 Text) to determine the parameters which greatly affect the model cycle length, then

**e.2** from the most sensitive parameters, we took some and vary them to make various cycle lengths, and

**e.3** applied the optimization code to determine timing of administration and dosage of exogenous hormones that result in anovulation.

**Table 2. Sum of squared residuals between the model output and Welt data, and model output peaks.**

| Hormone | Sum of squared residuals | Hormone peak | | |
|---|---|---|---|---|
| | | Hormone | Welt data | Model output |
| $E_2$ | 21840 (in $\text{pg}^2/\text{mL}^2$) | $E_2$ (in pg/mL) | 237 | 238 |
| $P_4$ | 183 (in $\text{ng}^2/\text{mL}^2$) | $P_4$ (in ng/mL) | 17.9 | 18.1 |
| $Inh$ | 66.6 (in $\text{IU}^2/\text{mL}^2$) | $Inh$ (in IU/mL) | 11.5 | 11.5 |
| $LH$ | 1579 (in $\text{IU}^2/\text{L}^2$) | $LH$ (in IU/L) | 123 | 123 |
| $FSH$ | 174 (in $\text{IU}^2/\text{L}^2$) | $FSH$ (in IU/L) | 19.6 | 20.4 |

## Results

This section presents the model output with and without administration of exogenous hormones. Sum of squared residuals between the model output without administration of exogenous hormones and Welt data, model output peak, and period are shown. In the administration of exogenous hormones, constant and time-varying doses are considered. Exogenous estrogen and progesterone monotherapies, and combination treatment of the two hormones are explored. For each type of therapy, minimum dose of exogenous estrogen and/or progesterone which leads to anovulation is determined. Optimal control theory is applied to investigate optimal time-varying doses.

### The normal cycle solution

Without the administration of exogenous hormones ($E_2^{\text{exo}}(t) = 0$ and $P_4^{\text{exo}}(t) = 0$), the estimated initial condition in Table 1 produces a unique stable periodic solution (called the normal cycle solution) to the menstrual cycle model. The cycle length is 28.05 days (also see Table 2). Local stability of the solution is affirmed by varying the initial conditions.

Fig 4 depicts the dynamics of the ovarian and pituitary hormones $E_2$, $P_4$, $Inh$, $LH$, and $FSH$ predicted by the model. It presents four cycles showing periodicity of the model output. The normal cycle solution (blue curve) exhibits hormone surges and dips and is a good estimate to the data extracted from Welt et al. [15] (red curve). Table 2 presents the sum of squared residuals between the model output and Welt data, and hormone output peaks.

### Administration of exogenous hormones

*Constant dosage.* Exogenous estrogen and/or progesterone inhibits pituitary and ovarian maximum hormone levels [37, 38]. To simulate the response to a constant dose of exogenous estrogen monotherapy, $E_2^{\text{exo}}(t) = 20$ pg/mL per day and $P_4^{\text{exo}}(t) = 0$ are used in Eqs (14) and (15) for 28 days. Similarly, the effect of a constant dose of exogenous progesterone monotherapy is obtained with $P_4^{\text{exo}}(t) = 1.4$ ng/mL per day and $E_2^{\text{exo}}(t) = 0$. Table 3 presents the percentage

**Table 3. Percentage decrease in model output peak with the administration of exogenous estrogen/progesterone.**

| Hormone | $E_2^{\text{exo}}(t) = 20$ pg/mL per day | | $P_4^{\text{exo}}(t) = 1.4$ ng/mL per day | |
|---|---|---|---|---|
| | Current model | Wright model | Current model | Wright model |
| $E_2$ (in pg/mL) | 21 | 16 | 51 | 81 |
| $P_4$ (in ng/mL) | 35 | 23 | 53 | 81 |
| $Inh$ (in IU/mL) | 33 | 22 | 55 | 82 |
| $LH$ (in IU/L) | 12 | 26 | 81 | 89 |
| $FSH$ (in IU/L) | 9 | 13 | 35 | 29 |

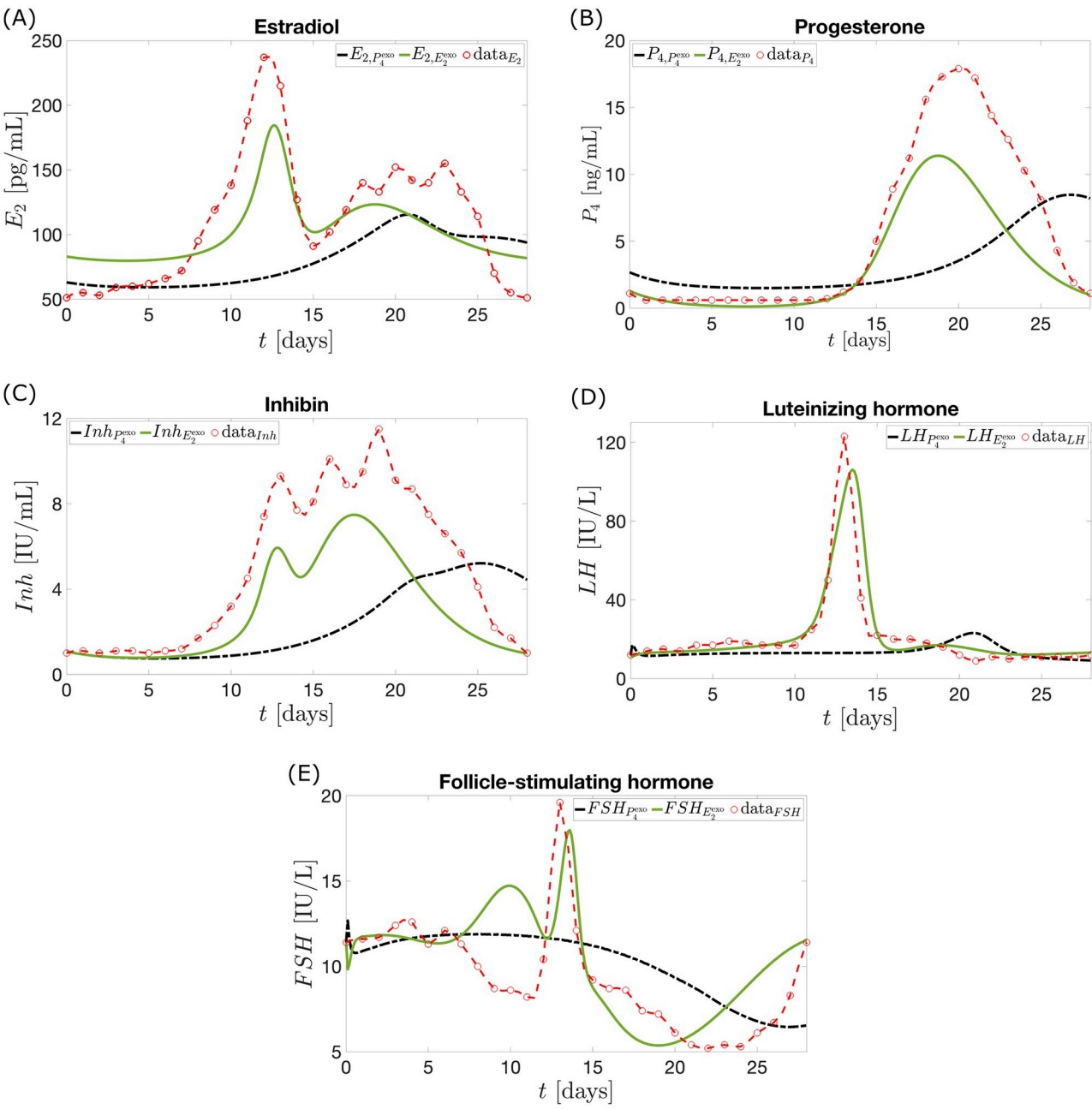

**Fig 5. Model output with constant dose of exogenous hormone.** $H_{P_4^{exo}}$ (black curve) or $H_{E_2^{exo}}$ (full green curve) is the hormone $H$ model output in case of 20 pg/mL per day of exogenous $E_2$ or 1.4 ng/mL per day of exogenous $P_4$ is administered for 28 days, respectively. The stipulated red curve represents the data for the 28-day normal cycle extracted from Welt et al. [15]. The addition of exogenous $E_2$ or $P_4$ reduces the peak of each of the five hormones.

decrease in model output peak compared to the hormone output peak of the Wright model of hormonal contraception [13]. The use of 20 pg/mL per day of estrogen or 1.4 ng/mL per day of progesterone results to the hormone profiles in Fig 5. Each of these amounts is insufficient to manifest anovulation because although reduced, the maximum $P_4$ value is still more than 5 ng/ml. To determine $E_2^{exo}(t)$ and $P_4^{exo}(t)$ values which block ovulation, we observe the $LH$ and $P_4$ model output as the dosages vary from 0 to 60 pg/mL per day on a 0.1 pg/mL-interval for

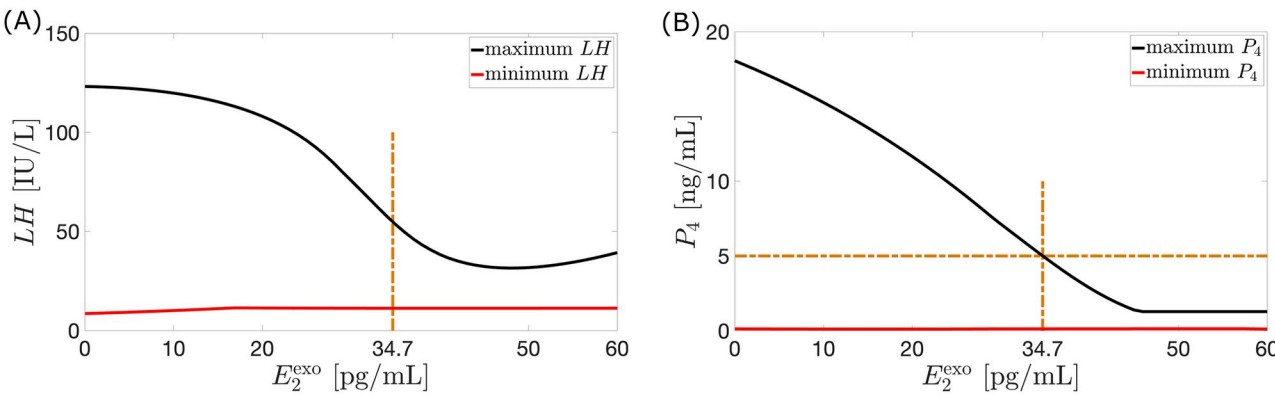

**Fig 6. Varying $E_2^{exo}$ dose.** The vertical axes in (A) and (B) present the maximum (full black curve) and minimum (full red curve) values over a 28-day cycle reached by $LH$ and $P_4$, respectively, when the corresponding amount of exogenous $E_2$ is given. Panel (B) is similar to (A) but shows that increasing dosage of $E_2^{exo}$ causes decreasing amplitude of the variation in $P_4$ level. Anovulation is attained when $E_2^{exo} > 34.7$ pg/mL.

$E_2^{exo}(t)$, and 0 to 4 ng/mL per day on a 0.1 ng/mL-interval for $P_4^{exo}(t)$. Figs 6 and 7 illustrate that increasing dosage leads to eliminating $LH$ surge and decreasing fluctuation in $P_4$ level. In the estrogen monotherapy, higher $E_2^{exo}(t)$ generates lower maximum $P_4$ and anovulation is attained when $E_2^{exo}(t) > 34.7$ pg/mL. In progesterone monotherapy, ovulation is suppressed between doses 3.1 ng/mL and 3.7 ng/mL. The small window of anovulation is due to $P_4^{exo}(t)$ in Eq (15), and the linear inhibitory terms $P_4(t)/w$ and $P_4(t)/q$ in Eqs (3) and (5). Because the inhibitory term $P_4(t)/w$ is linear, to bring $P_4(t) < 5$ ng/mL the amount of $P_4(t)$ must provide strong inhibition of $FSH$. This is attained by increasing $P_4^{exo}(t)$ in Eq (15). A high amount of $P_4^{exo}(t)$ also enhances inhibition of the linear term $P_4(t)/q$ on $RcF(t)$, which in turn suppresses further $Lut_3(t)$ and $Lut_4(t)$. However, because of the inclusion of $P_4^{exo}(t)$ in Eq (15), increasing amount of $P_4^{exo}(t)$ also results to raising $P_4(t)$ toward values more than 5 ng/mL, which are ovulatory levels.

As in [13], an anovulatory effect of a high dosage of exogenous estrogen or progesterone can be achieved by using a combination of the two hormones. For instance, monotreatment by $P_4^{exo}(t)$ between 3.1 ng/mL per day and 3.7 ng/mL per day or $E_2^{exo}(t)$ greater than 34.7 pg/mL per day for the entire 28-day cycle induces anovulation (see Figs 6(B) and 7(B)). Fig 8(A)

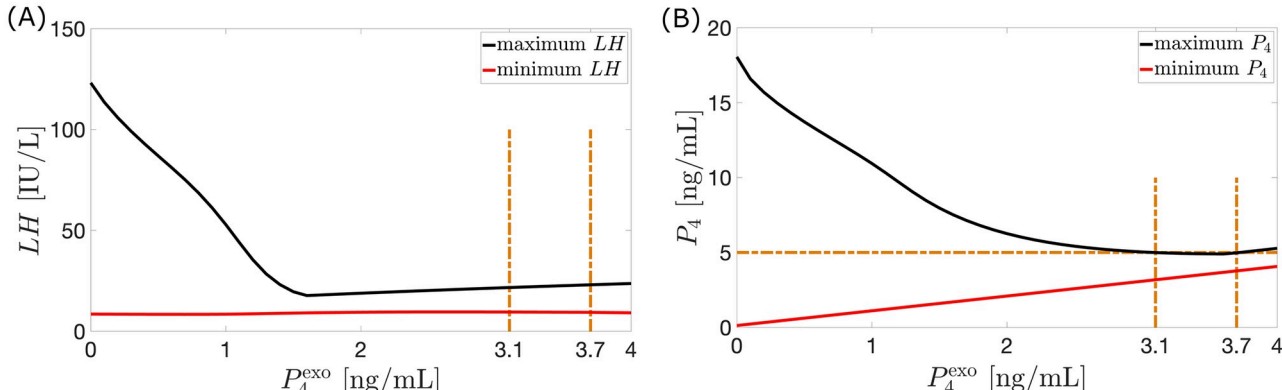

**Fig 7. Varying $P_4^{exo}$ dose.** Shown in (A) and (B) are the maximum (full black curve) and minimum (full red curve) values attained by $LH$ and $P_4$ over a 28-day cycle resulting from the administration of the corresponding dosage of $P_4^{exo}$. Panel (B) illustrates diminishing fluctuation in $P_4$ value. Anovulation is achieved between $P_4^{exo} = 3.1$ ng/mL and $P_4^{exo} = 3.7$ ng/mL.

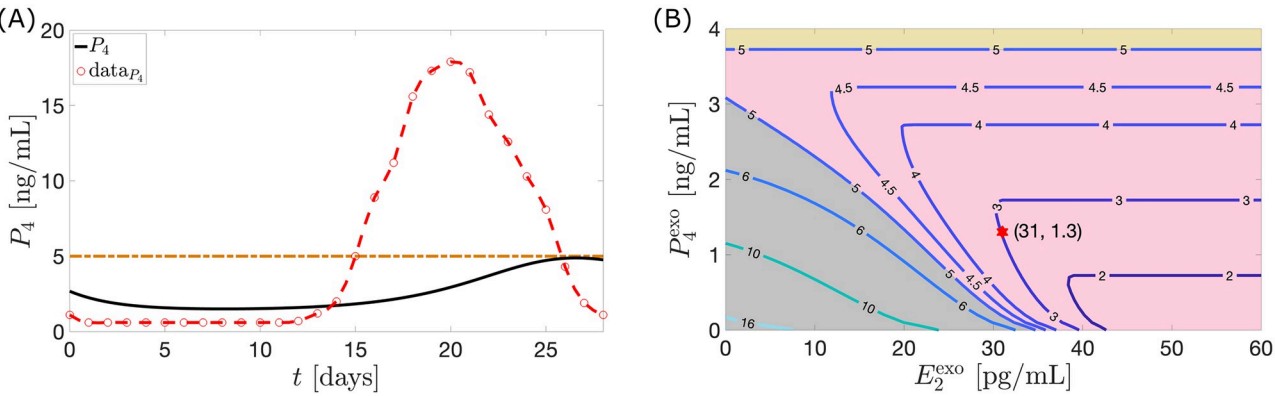

**Fig 8. Constant dose combination therapy.** (A) The black curve represents the $P_4$ model output for a combination treatment with $P_4^{\mathrm{exo}} = 1.4$ ng/mL per day and $E_2^{\mathrm{exo}} = 20$ pg/mL per day administered for 28 days. The red circles with the interpolated curve represents the 28-day normal cycle hormone data extracted from Welt et al. [15]. The treatment suppresses the $P_4$ concentration achieving anovulatory state. In (B) the curves composed of points $(E_2^{\mathrm{exo}}, P_4^{\mathrm{exo}})$, correspond to a combined dosage of $E_2^{\mathrm{exo}}$ and $P_4^{\mathrm{exo}}$ resulting in a $P_4$ maximum value of $k$ ng/mL. Anovulation is attained when $k < 5$. The yellow region ($P_4^{\mathrm{exo}} > 3.7$ ng/mL) corresponds to ovulation. On the lower left portion, an almost straight line with slope $-0.1$ separates the regions of ovulation (in gray) and anovulation (in pink).

shows that alternatively, anovulation can be obtained by a combination treatment with $P_4^{\mathrm{exo}}(t) = 1.4$ ng/mL per day and $E_2^{\mathrm{exo}}(t) = 20$ pg/mL per day. Recall that the monotreatment with $E_2^{\mathrm{exo}}(t) = 20$ pg/mL per day or $P_4^{\mathrm{exo}}(t) = 1.4$ ng/mL per day does not prevent ovulation (see Fig 5(B)). Other combination treatments which block ovulation are depicted in Fig 8(B).

Fig 8(B) shows contour plots of maximum $P_4$ levels over a cycle for various combination treatments of $E_2^{\mathrm{exo}}(t)$ and $P_4^{\mathrm{exo}}(t)$. For instance, the point (31, 1.3) on the contour labeled 3 signifies that an infusion of a combination of $E_2^{\mathrm{exo}}(t) = 31$ pg/mL per day and $P_4^{\mathrm{exo}}(t) = 1.3$ ng/mL per day yields a maximum $P_4$ level of 3 ng/mL. The region of anovulation (in pink) is bounded above and on the left by the curves $k = 5$. The left boundary is approximately a straight line with slope $-0.1$. Thus, to stay on this boundary a reduction of 1 pg/mL per day of $E_2^{\mathrm{exo}}(t)$ needs to be counteracted by an increase of approximately 0.1 ng/mL per day of $P_4^{\mathrm{exo}}(t)$.

To determine optimal constant dosage resulting in anovulation, $P_4$ output peaks are recorded for each combination of $E_2^{\mathrm{exo}}(t)$ and $P_4^{\mathrm{exo}}(t)$ dose. For the estrogen monotherapy, the dosage of $E_2^{\mathrm{exo}}$ combined with $P_4^{\mathrm{exo}} = 0$ (dosage of $P_4^{\mathrm{exo}}$ combined with $E_2^{\mathrm{exo}} = 0$ for progesterone monotherapy) resulting in $P_4$ peak of 4.99 ng/mL is taken as the optimum constant dosage. In the combined therapy, the optimum constant dosage is the nonzero dosage of $E_2^{\mathrm{exo}}$ combined with nonzero $P_4^{\mathrm{exo}}$ resulting in $P_4$ peak of 4.99 ng/mL.

### Optimal nonconstant dosage

This section uses optimal control to determine optimal time-varying doses that induce anovulation. To explore mono and combination treatments, three different cases of the objective function are examined.

**Case 1**. **Exogenous estrogen as monotherapy**. Let $u_2(t) = 0$. For $a_1 = 0.4$ $\mu$g/mL then the control $u_1(t)$ which minimizes the objective function is illustrated in Fig 9(A).

In Fig 9, the steep rise in the dosage of exogenous estrogen in the optimal control inhibits strongly the release of *FSH* in the bloodstream via Eq (3). This causes the *FSH* levels around the time of $u_1$ surge to plunge. Consequently, a low mass of *RcF* (see Fig 10) is produced via Eq (5). The underdeveloped follicles then produce lower $E_2$, which causes a decrease in *LH* production via Eq (1). Without an *LH* surge, ovulation does not occur. This implies the value of $P_4$ to be lowered via Eq (14).

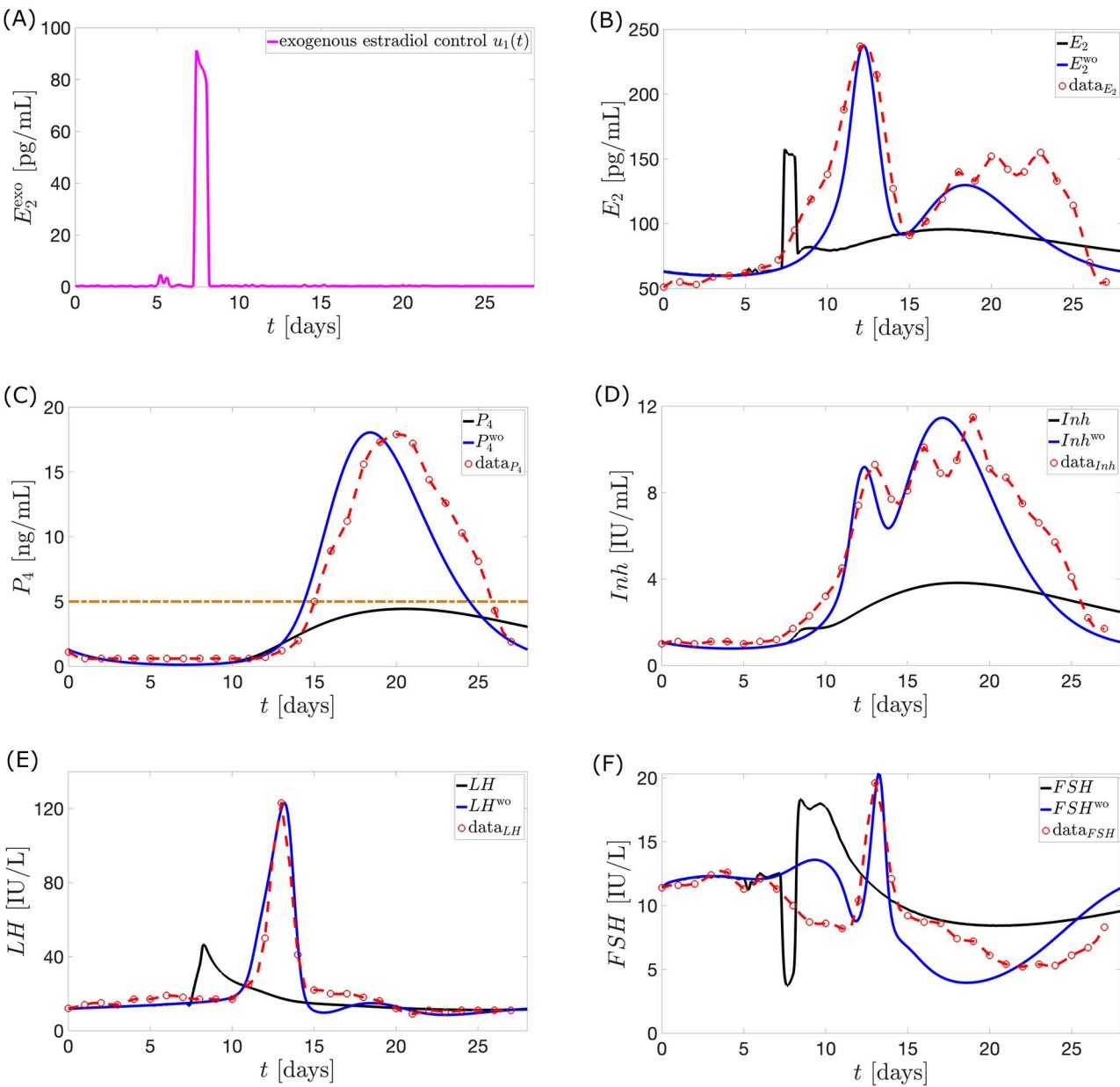

**Fig 9. Model output with application of optimal control $u_1$.** The full black curve is the model output when the optimal control $u_1$ (magenta in panel (A)) is applied. $H^{wo}$ (full blue curve) denotes the hormone model output without the influence of $u_1$. This is the normal cycle solution. The Welt data for a normal cycle is presented in red circles with the interpolated curve. The maximum $P_4$ value is 4.43 ng/mL.

The large dosage of $E_2^{exo}(t)$ given in the mid-follicular phase is effective in preventing the ovulatory level of $E_2$ in the late follicular phase. The schedule of administration agrees with [52] that estrogen treatment started before the 10th day of the menstrual cycle can result in anovulation.

Increasing the value of $a_1$ penalizes $u_1$ dose more strongly. This reduces $u_1$ dose further but increases deviation of $P_4$ from 4 ng/mL.

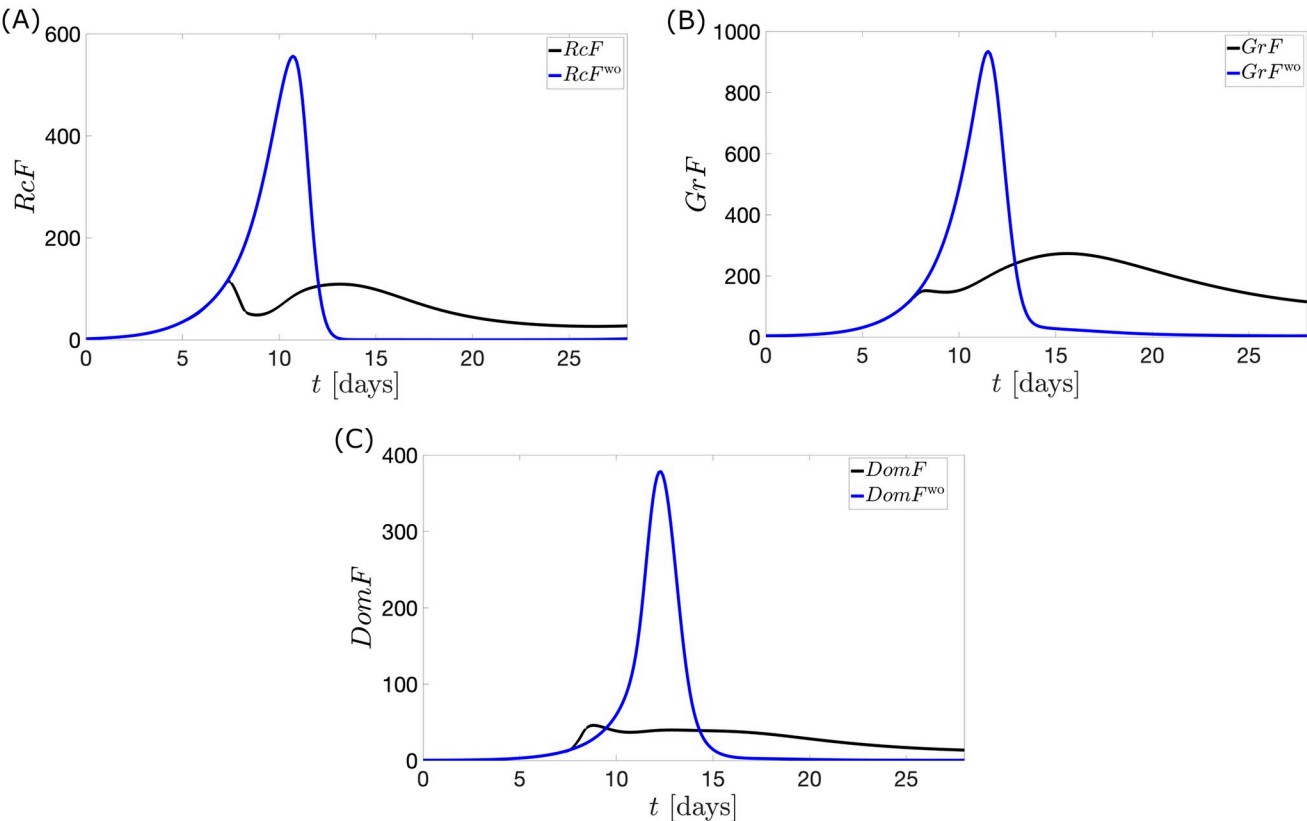

**Fig 10. Follicular mass with application of optimal control $u_1$.** The full black curve describes the follicular mass when the optimal control $u_1$ is applied. The full blue curve shows the follicular mass without the application of $u_1$. In (A), the steep decline in *RcF* is evident on the interval of *FSH* level drop in Fig 9(F). The inhibition of *RcF* subsequently contributes to the reduced development of *GrF* and *DomF* in panels (B) and (C).

**Case 2**. **Exogenous progesterone as monotherapy**. Assume $u_1(t) = 0$ and $a_2 = 0.07$ mL$^2$/ng$^2$. Anovulatory hormonal levels result from a continuous suppression of *FSH* levels (see Fig 11 (F)), in contrast to the sudden dip in Fig 9(F).

The administration of $u_2$ starting from the first day of the cycle does not permit *FSH* to reach its maximum value due to the low synthesis in the pituitary via Eq (3). The low level of *FSH* in the follicular phase and the additional inhibition by $P_4$ via Eq (5) hinder follicular growth. Consequently, $E_2$ levels far lower than normal are attained via Eq (14). Nonoccurrence of *LH* surge follow via Eq (1).

The optimal control $u_2$ suggests that a maximum dosage be given before the time in the normal menstrual cycle when $P_4$ peaks. *Inh* prevents *FSH* synthesis via Eq (3) but anovulation results to decreased *Inh* levels via Eq (16). Thus, the mathematical model causes $u_2$ to still have high doses after day 14 in order to continue suppression of *FSH* production by compensating for the reduced inhibition by *Inh*.

Similar to Case 1, increasing value of $a_2$ puts more effort in minimizing $u_2$ dose, increasing deviation of $P_4$ from 4 ng/mL.

**Case 3**. **Administration of combined exogenous estrogen and progesterone**. Allowing $u_1(t) \neq 0$, $u_2(t) \neq 0$, $a_1 = 0.4$ μg/mL and $a_2 = 0.7$ mL$^2$/ng$^2$, then the optimal controls $u_1$ and $u_2$ illustrated in Fig 12(A) yield the dynamics in Fig 12(B) to 12(F).

The weights attached to the $u_1$ and $u_2$ terms result to smaller percentage decrease of $u_1$ peak from Case 1 compared to the percentage decrease of $u_2$ peak from Case 2. This leads to a

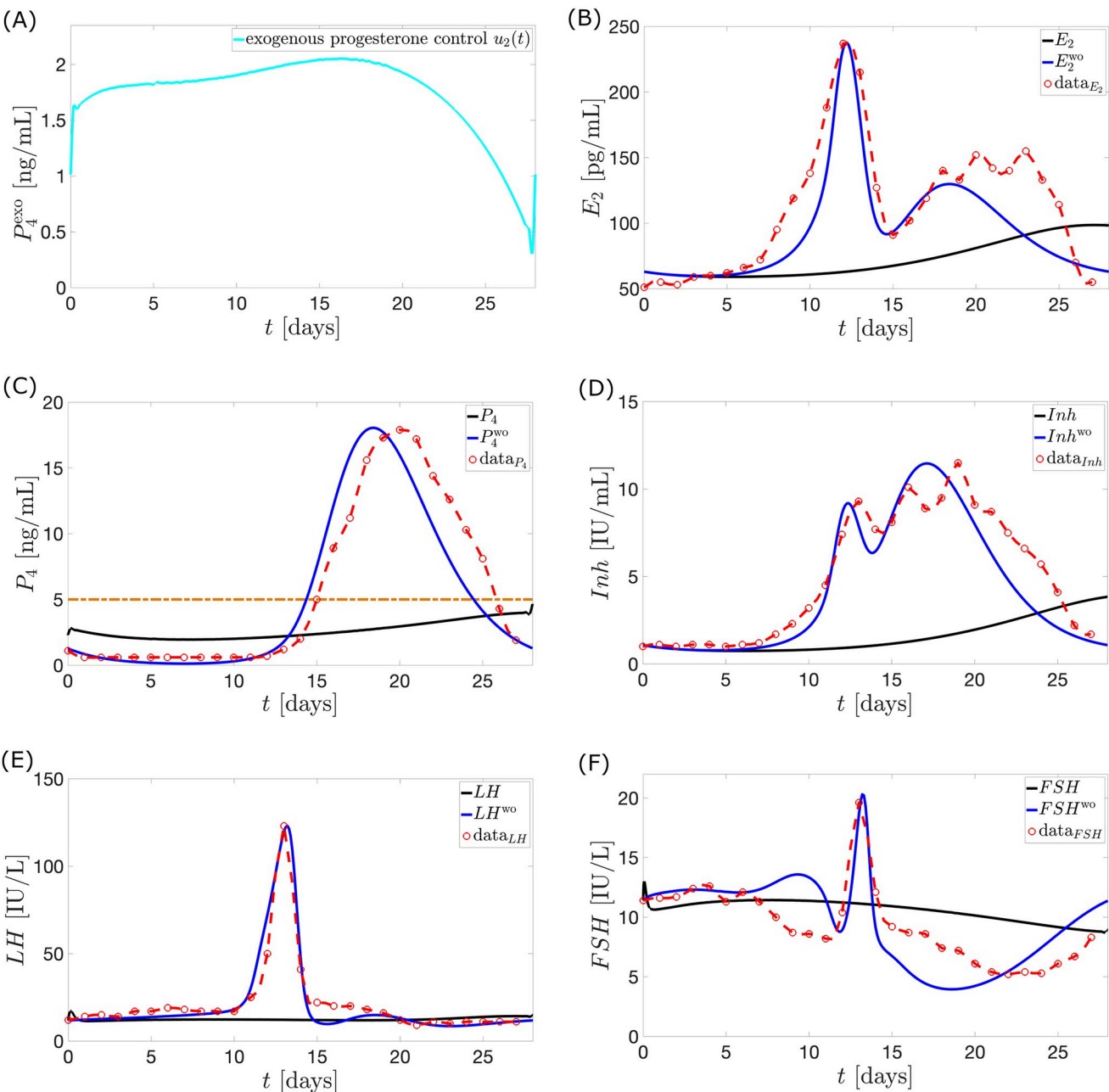

**Fig 11. Model output with application of optimal control $u_2$.** The full black curve is the model output when the optimal control $u_2$ (cyan in panel (A)) is used. The normal cycle solution $H^{\text{wo}}$ (full blue curve) is the hormone model output when $u_2$ is not administered. The red circles with the interpolated curve denote the Welt data for a normal cycle. $P_4$ reaches a maximum value of approximately 4.66 ng/mL.

greater influence of the optimal control $u_1$ on the menstrual cycle. Fig 12 presents hormone profiles resembling the ones shown in Fig 9. The weight $a_2$ may be decreased if the intention is to diminish the impact of $E_2$.

Not only does the combination therapy utilize lower doses of exogenous estrogen and progesterone, it also allows the administration of $u_1$ to commence in a later follicular stage. Our simulation shows that the sole administration of estrogen (see Fig 9(A)) blocks ovulation if it is

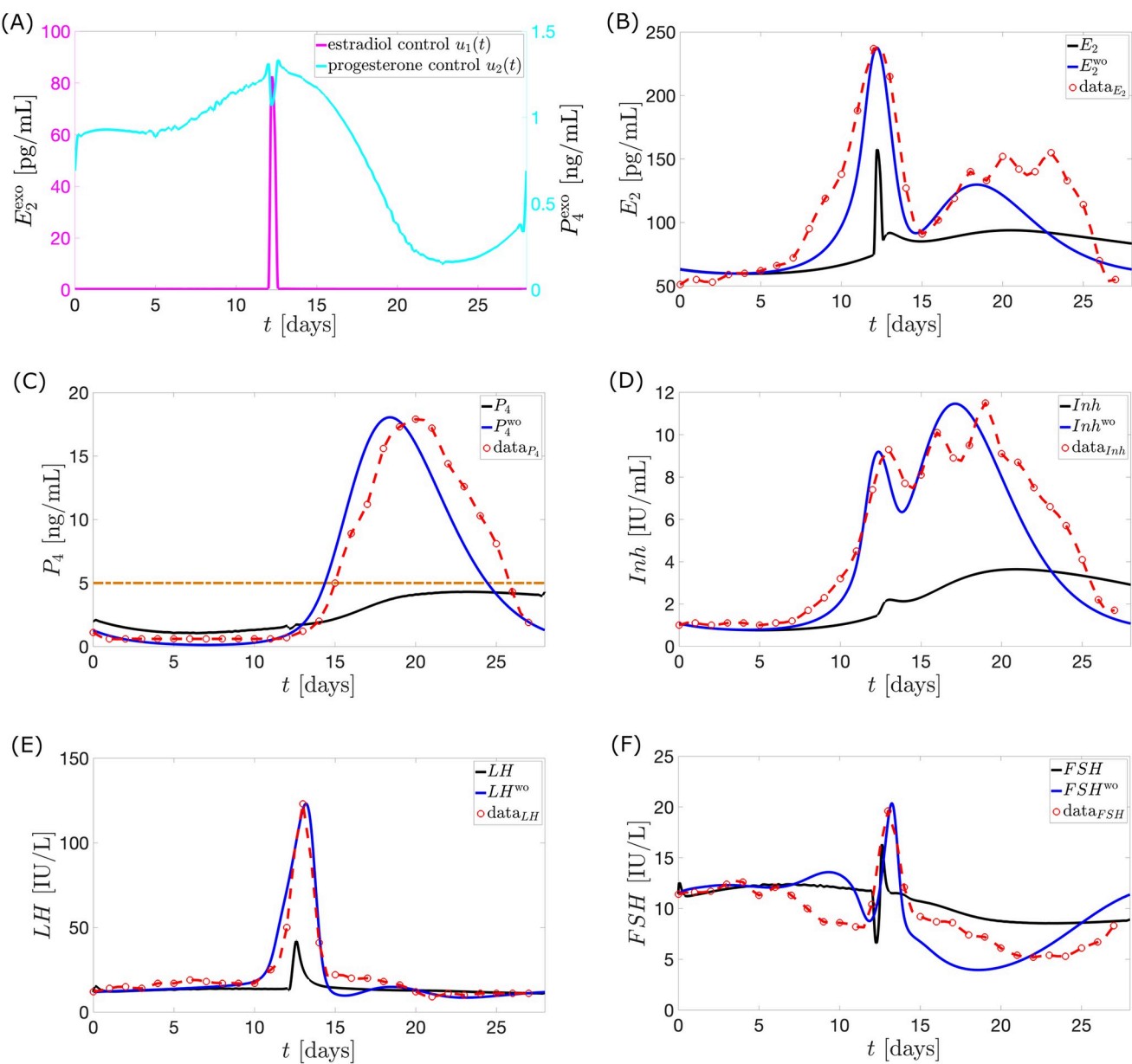

**Fig 12. Model output with application of optimal controls $u_1$ and $u_2$.** The full black curve is the model output when the optimal controls $u_1$ and $u_2$ (magenta and cyan in panel (A)) are administered. The normal cycle solution $H^{\text{wo}}$ (full blue curve) is the hormone model output when $u_1$ and $u_2$ are not applied. The red circles with the interpolated curve denote the Welt data for a normal cycle. The maximum $P_4$ concentration is approximately 4.31 ng/mL.

done prior to the 10th day of the cycle. Interestingly, the combination therapy (see Fig 12 (A)) suggests that time-varying doses of estrogen and progesterone given simultaneously from the start to the end of the 28-day period, only requires a surge in estrogen dose around the 12th day of the cycle (a delayed administration compared to the estrogen monotherapy). The late administration of $u_1$ is possibly compensated by the inhibition provided by exogenous $P_4$ in the early follicular phase. The surge in the $u_1$ around $t = 12.2$ days allowed a dip in $u_2$. The combination of the $u_1$ peak dose and the $u_2$ dose at this time maintains anovulation. Unlike in Case 1, the dosage and timing of administration of $u_1$ does not suffice to

keep anovulatory $P_4$ level until the end of cycle thus, use of increasing $u_2$ is needed in the late luteal phase.

## Discussion

With the rapid development of new implants and injections providing continuous administration there is great potential to implement new treatment scheme minimizing dose. This study employs optimal control to a modification of the model in Margolskee et al. [11] to determine the optimal time-varying dose of exogenous estrogen and/or progesterone that induce anovulation. Biologically, the period length vary within people and between people. This work does not account for this variation, though it can be introduced by varying model parameters or embedding the model in a stochastic framework. This limitation was introduced to generate a simple model that show that contraception can be obtained by manipulating certain parts of the menstrual cycle. The parameter estimation employed yields a model output that predicts the data extracted from Welt et al. [15]. An improvement to previous mathematical models, hormone output peaks are close to the data peaks (see Table 2). This is essential since an anovulatory cycle is determined from the reduction of the normal maximum $P_4$ level to less than 5 ng/mL and the lack of $LH$ surge. The cycle length is also close to the 28-day period of the Welt data. This makes the model beneficial in future studies investigating the effect of exogenous hormones on cycle length.

Exogenous estrogen and/or progesterone inhibits pituitary and ovarian maximum hormone levels [37, 38]. In the model, the administration of exogenous estrogen and/or progesterone caused a reduction in maximum hormonal values (see Fig 5). Such effect is also produced by the hormonal contraception model in Wright et al. [13]. With $E_2^{\text{exo}}(t) = 20$ pg/mL per day, the current model suppresses more the peaks of three of the five hormones but with $P_4^{\text{exo}}(t) = 1.4$ ng/mL per day, the Wright model reduces more the peaks of four of the five hormones (see Table 3). The model in Wright et al. [13] provides greater repression by exogenous $P_4$ because it uses a nonlinear term to inhibit $RcF$ growth and an additional equation depicting upregulation of $P_4$ by $E_2$, boosting the contraceptive effect of $P_4$. We opted for the linear inhibitory term $P_4(t)/q$ and fewer additions to the Margolskee model [11] to keep the current model simple, reducing the computation time in running our optimization code. This is because the numerical method used in this study, control parameterization using MATLAB functions dde23 and fmincon, though easy to implement is not cost effective. Computation time ranges from few hours to several days depending on the proximity of initial guess to the optimal solution. Table D in S1 Text illustrates some examples.

To our knowledge there has been no study applying optimal control theory on the menstrual cycle model. Optimal control could provide drug administration scheme which greatly enhances contraception outcome by significantly minimizing risks associated with high doses such as venous thromboembolism and myocardial infarction [3, 30–32]. The study by Gu et al. [53] showed that compared to constant-dose administration, optimal control results could substantially improve HIV treatment. Results of our work similarly suggest the significant advantage of optimal time-varying doses.

In estrogen monotherapy, the minimum constant dosage of estrogen over 28 days resulting in an anovulation is (34.73 pg/mL) × 28 = 972.44 pg/mL (see Fig 13(A)). This dosage lowers the maximum $P_4$ level to 4.99 ng/mL. The administration of the optimal control $u_1$ is able to bring down the maximum $P_4$ level to 4.43 ng/mL (i.e., anovulation is achieved) with only a total dosage (area under the curve or AUC) of 77.76 pg/mL (see Fig 13(B)). A dosage of 894.68 pg/mL (about 92% of minimum total constant dosage) would be saved if $u_1$ is used to induce anovulation.

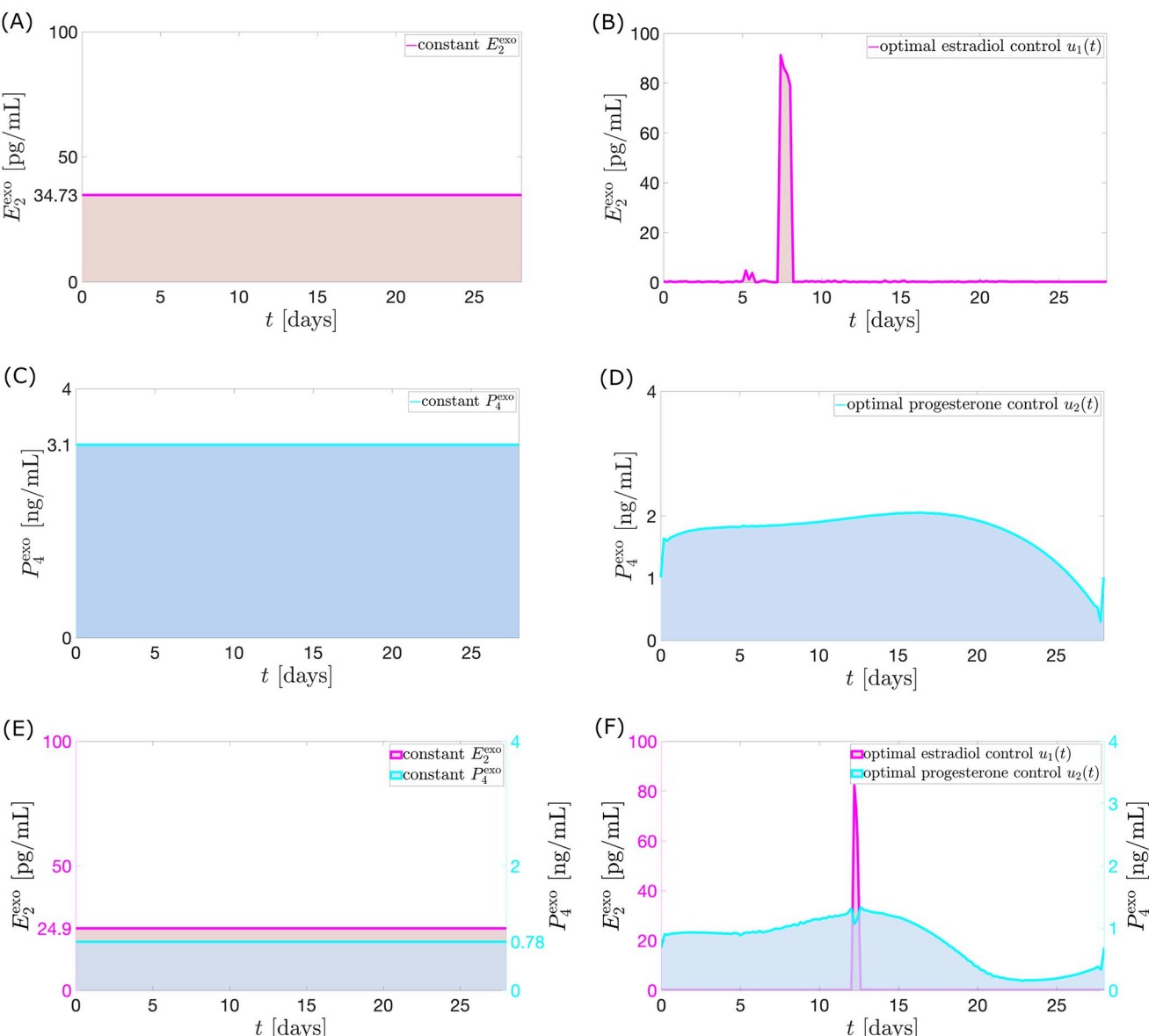

**Fig 13. Constant dosage and nonconstant dosage comparison.** The shaded regions in Panels (A), (C), and (E) indicate the minimum total constant dosage of exogenous estrogen and/or progesterone over 28 days that lowers maximum $P_4$ concentration to 4.99 ng/mL. The shaded region below $u_1$ (area under the curve or AUC) in Panel (B) is the total nonconstant dosage of exogenous $E_2$ which suppresses the $P_4$ level to 4.43 ng/mL, a reduction by about 92% of the total dosage in (A). Panel (D) illustrates the total nonconstant dosage of exogenous $P_4$ that reduces maximum $P_4$ to 4.66 ng/mL, a reduction by about 43% of the total dosage in (C). Panel (F) shows the combined nonconstant doses of exogenous $E_2$ and $P_4$ that gives a maximum $P_4$ level of 4.31 ng/mL.

A constant intravenous infusion of a low dose of estradiol result in complete shut down of the GnRH pulse generator on female rhesus monkeys for several days or weeks [54, 55]. Although endometrial bleeding is rare in this species, a constant administration may induce a growth of female human endometrium exposing to hyperplasia and bleeding. It justifies the need for an open window without estrogen to allow endometrial bleeding. The optimization result for the time-varying administration of estrogen suggests a dosing regimen generating an estrogen-free window. As is the case with new drug regimen, clinical studies would further

assess the effects of restraining the administration of exogenous estrogen to shorter periods and varying regimens.

In the progesterone monotherapy, $u_2$ is able to bring down the maximum $P_4$ level in 28 days to 4.66 ng/mL with only a total dosage (AUC) of 48.84 ng/mL (see Fig 13(D)). A constant administration would require 3.1 ng/mL per day, a total dosage of (3.1 ng/mL) $\times$ 28 = 86.8 ng/mL to lower the maximum $P_4$ to 4.99 ng/mL (see Fig 13(C)). A dosage of 37.96 ng/mL (about 43% of the total dosage for constant administration) would be saved if $u_2$ is employed.

Exogenous progesterone like progestin may affect the hypothalamo-pituitary ovarian axis differently compared to endogenous progesterone. The current model shows a principle of contraception with exogenous administration of progesterone. If the influence of a specific exogenous progesterone is represented, the current model should be coupled with a pharmacokinetics model that describes the details of the drug.

Now in the combination therapy, the total dosage given by $u_1$ (AUC$u_1$) is 35.58 pg/mL while that of $u_2$ (AUC$u_2$) is 21.67 ng/mL (see Fig 13(F)). If AUC$u_2$ is taken and spread out constantly in 28 days, then $P_4^{\text{exo}}(t) = 0.78$ ng/mL (see Fig 13(E)). Consider the least amount of $E_2^{\text{exo}}(t)$ administered constantly in combination with this $P_4^{\text{exo}}(t)$ that results to anovulation. Guided by the contour plot (see Fig 8(B)), $E_2^{\text{exo}}(t) = 24.9$ pg/mL suppresses maximum $P_4$ to 4.99 ng/mL. The difference between the total exogenous estrogen dosage between the constant and nonconstant administration is (24.9 pg/mL) $\times$ 28 $-$ AUC$u_1$ = 661.62 pg/mL. Hence about 94.89% of the total $E_2^{\text{exo}}$ dosage for constant administration would be saved if the combination of $u_1$ and $u_2$ is taken. On the other hand, if the total dosage (AUC$u_1$) given by $u_1$ is spread out constantly in 28 days, then $E_2^{\text{exo}}(t) = 1.27$ pg/mL. The contour plot (see Fig 8(B)) implies that the least amount of $P_4^{\text{exo}}(t)$ which can be given constantly in combination with $E_2^{\text{exo}}(t)$ to decrease maximum $P_4$ to 4.99 ng/mL is $P_4^{\text{exo}}(t) = 3$ ng/mL. The difference between the total $P_4^{\text{exo}}(t)$ dosage between the constant and nonconstant administration is (3 ng/mL) $\times$ 28 $-$ AUC$u_2$ = 62.33 ng/mL. Hence about 74.20% of the total $P_4^{\text{exo}}$ dosage for constant administration would be saved if the combination of $u_1$ and $u_2$ is used to inhibit ovulation. A summary of the results of the optimal control strategies is shown in Table 4.

The optimal control results provide the best timing of administration since an earlier or delayed application of $u_1$ and/or $u_2$ yields higher $P_4$ level. For instance, if the large-dose portion of $u_1$ is applied in equal intervals from day 35 to day 280 (see Fig 14, anovulation is no longer induced beginning on the 5th 28-day period (i.e. from day 140 to day 280). This is because the application of $u_1$ in the preceding period changes the dynamics of the menstrual cycle in the succeeding days. One of these changes is the cycle length. Now, because the control $u_1$ must be administered at a time before $E_2$ surge (when $u_1$ is not applied), the timing of administration in the next cycles must also be changed to continuously suppress ovulation. Sometimes resulting cycle length is less than 28 days so only a portion of $u_1$ will be applied. In addition, because lower levels of $u_1$ are negligible compared to the higher levels, we applied only the large-dose portion of $u_1$ from day 35 to day 280 (see Fig 14). The administration is done when $E_2$ level is increasing and reaches 75 pg/mL. We are currently conducting further investigation to

**Table 4. Maximum progesterone level throughout a menstrual cycle caused by the indicated total dose of the optimal exogenous hormone.**

| Treatment regimen | Total dose | maximum $P_4$ level |
|---|---|---|
| estrogen monotherapy | 77.8 pg/mL | 4.43 ng/mL |
| progesterone monotherapy | 48.8 ng/mL | 4.66 ng/mL |
| combination therapy | estrogen = 35.6 pg/mL<br>progesterone = 21.7 ng/mL | 4.31 ng/mL |

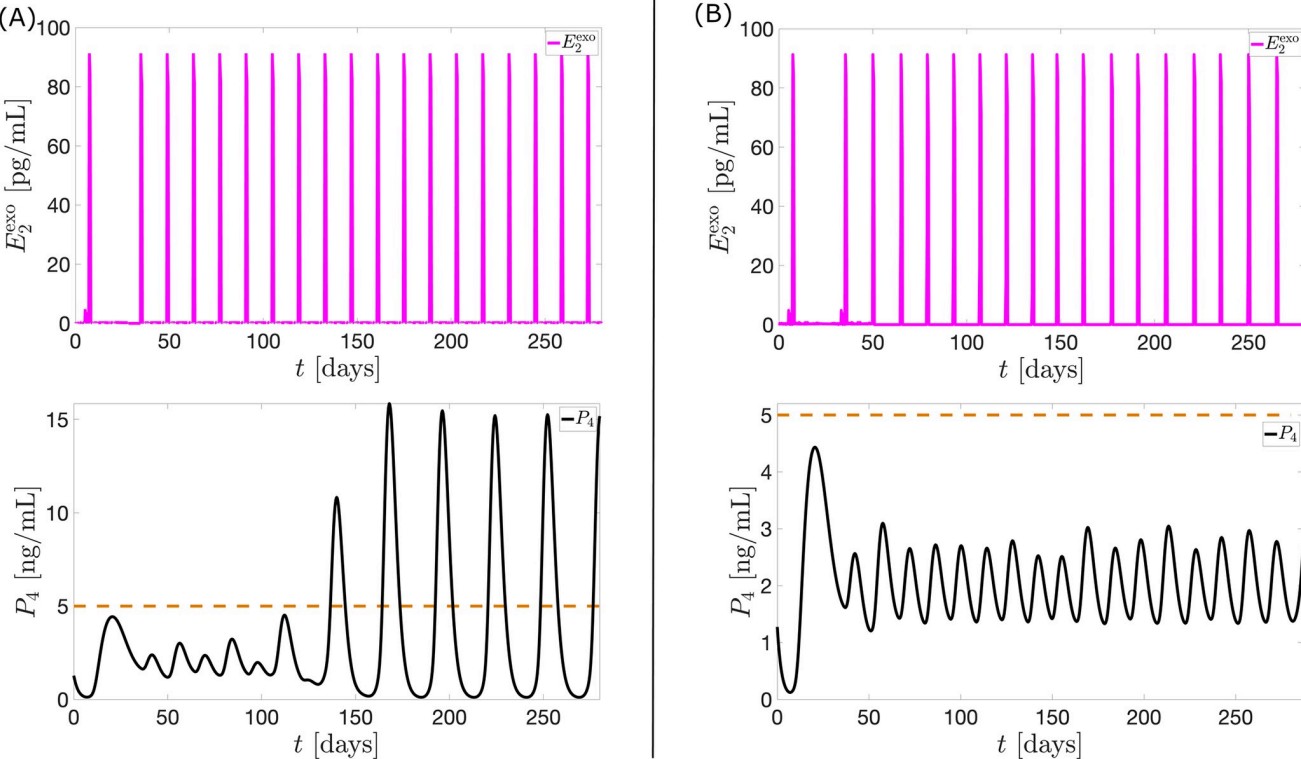

**Fig 14. Multiple application of optimal control $u_1$.** The black curve is the model output when multiple $u_1$ (magenta curve) is applied. In panel (A), the application of $u_1$ from day 35 to day 280 in equal intervals is unable to sustain anovulation. Panel (B) shows a scheme for administration of multiple $u_1$ which continuously blocks ovulation.

determine the highest $E_2$ hormone level where administration of $u_1$ would commence to still inhibit ovulation. Note that although $u_1$ is applied eighteen times (in unequal intervals) from day 0 to day 280, the total dosage is still significantly lower than the total constant dosage of exogenous $E_2$ which induces anovulation. The dosing regimen presented, where administration is triggered by a specific biomarker, here the 75 pg/mL-$E_2$ level, offers insights on construction of timed devices that give contraception at certain parts of the menstrual cycle. For instance, contraceptive dose may be given relative to $E_2$ level. This is analogous to the study where $LH$ levels are used as indicator for the time of antagonist administration in GnRH antagonist protocols [56]. So even if some patients' cycle length vary a little bit from month to month, their $E_2$ level can be measured to know when to make a contraceptive device spike.

Result of the sensitivity analysis performed on the model (see Fig A in S1 Text) shows parameters which largely affect model output. To explore adaptability of our control method to different menstrual cycle conditions, we perturb some of the most sensitive parameters to generate model output with different cycle lengths and peaks. Because of its biological significance, we perturbed the third most sensitive parameter $km_{LH}$. This parameter is the $E_2$ value at half saturation, it signals strong stimulation in the production of $E_2$ necessary for ovulation. With $km_{LH}$ equal to 115 pg/mL and 160 pg/mL, we yield model outputs of cycle length 26.92 days and 29.08 days, respectively. Applying the code for optimal nonconstant estrogen monotherapy gives the optimal control $u_1$ profile in Fig 15(B). The effect of the control in Fig 15(B) on the $P_4$ level is illustrated in Fig 15(C). These figures show how the optimal control dosage and timing of administration adjust to the varied cycle types.

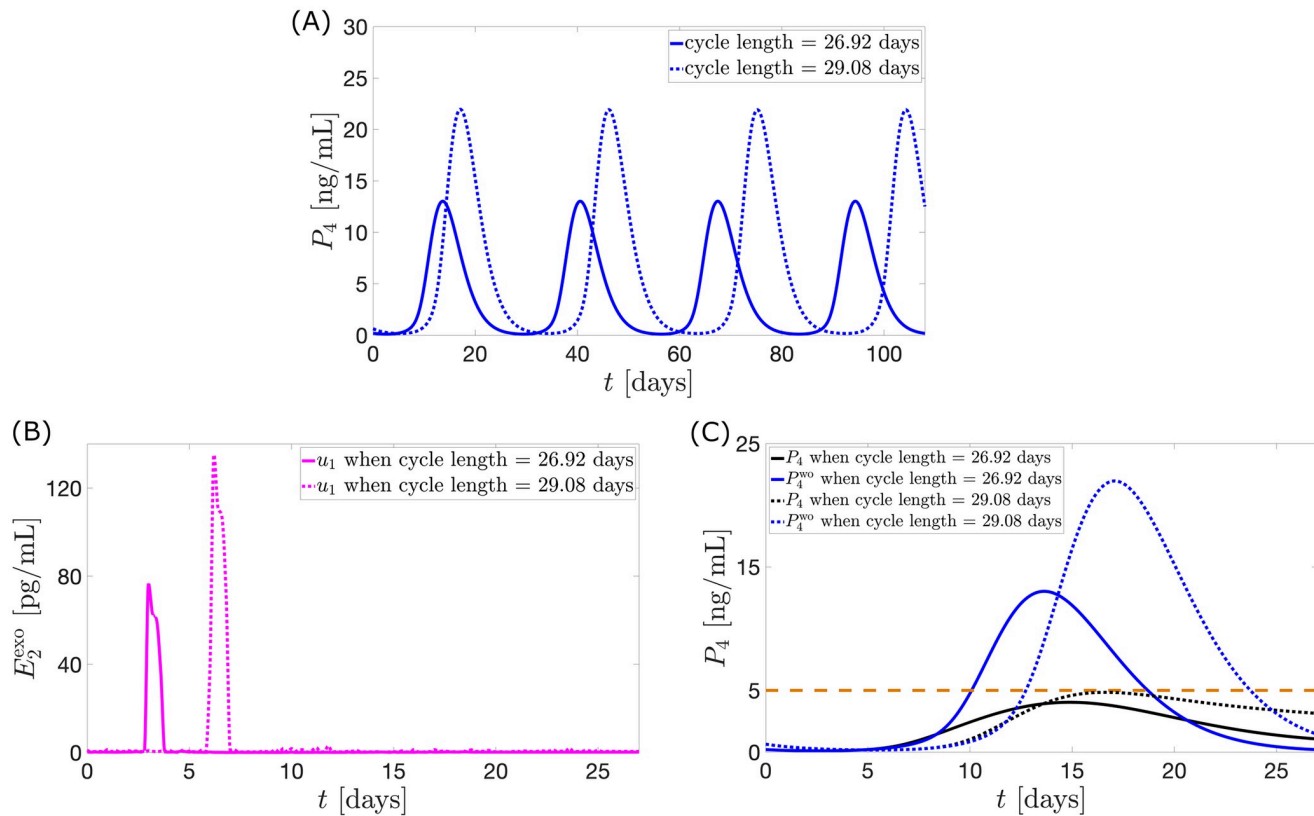

**Fig 15. Optimal control $u_1$ on different cycle lengths.** Panel (A) shows model output curves with different cycle lengths. Panel (B) presents the optimal control $u_1$ obtained by applying the objective function for estrogen monotherapy. The effect of $u_1$ administration on $P_4$ peak is described in Panel (C).

The methodology utilized in this study can be adapted to varied cycle conditions as discussed but we recognize some limitations. For instance, several studies have reported that menstrual timing is not precisely known in all women [57–60]. A prior knowledge of the menstrual timing can effectively identify appropriate administration schedule of hormonal contraception. The reproductive function in women is a very complex multiscale dynamical system highly dependent on both endogenous and exogenous hormones thus, the model developed in this study does not capture all factors involved with contraception. Rather, the model serves as a first step in using mathematical modeling to study the transition to a contraceptive state. When more data on individual hormone variations become available, the approach used here can be extended to use these. Another direction is to couple the model with a pharmacokinetics model to obtain patient-specific model to investigate the effects of specific contraceptives on individual menstrual cycle conditions. This provides avenue to examine further the complex multiscale effects of the factors included in silico. While at present, obtaining daily blood profile could be challenging both for financial and practical reasons, this study may motivate future development of advanced methodologies and technologies in data collection.

## Conclusion

This study employs a menstrual cycle model which correctly predicts the pituitary and ovarian levels throughout a normal cycle and reflects the decrease in maximum hormone levels caused by exogenous estrogen and/or progesterone. Optimal control results show that a significant reduction in the dosage of exogenous estrogen and/or progesterone may induce anovulation.

Furthermore, combination therapy lower doses even more. Simulations also show the effectivity of the administration of exogenous estrogen in the mid follicular phase.

Through the years, the reduction of exogenous estrogen and progesterone doses in contraceptives is done to decrease risks for adverse effects such as thrombosis and myocardial infarction. With the advent of fully automated hormone-delivery device like the intravaginal prototype device for cattles [61], continuous hormone administration to significantly reduce exogenous hormone dosages in humans is of interest. Hence, results presented in this paper may give clinicians guidance on conducting experiments about optimal treatment regimen causing anovulation. Because the model output cycle length of 28.05 days is a good approximation to the 28-day data cycle length, the model may also be used to explore how treatments change the period of the menstrual cycle. In future studies, researchers should consider stochasticity in the model to investigate within- and between- women's variabilities and couple the current model with a pharmacokinetics model to take into account the exact nature and metabolisms of administered hormones, allowing investigation into effects of specific drugs. Early optimal control results in this study exhibit noticeable oscillations. By several adjustments to the objective function, we were able to smoothen the control. To further lessen fluctuations in the optimization outcome, usage of other numerical techniques like polyhedral active set algorithm or introduction of dynamic equation to the objective function may be explored. Our work suggests a method of obtaining an optimal regimen for administration of exogenous hormone/s over one cycle. In the future when more cost-effective optimization scheme is accessible, one may build on this process to investigate dosing schemes for multiple cycles. Currently, the process applied over one cycle may be repeated for every cycle. It is also possible to use the result over one cycle to continually suppress ovulation over multiple cycles. Additionally, because there is a biomarker for level of hormones like $E_2$, $P_4$, and $LH$, the results presented here give insights on construction of timed devices that give contraception at certain parts of the menstrual cycle.

## Supporting information

**S1 Text. Supplementary information file.** This file presents the data used in our study, model parameters, standard deviation of model parameters, result of the sensitivity analysis performed on the model, effect of various weights on the optimal control results, optimal control results from different forms of the objective function, and computation time and optimal cost for the optimal control simulations.
(PDF)

## Acknowledgments

We thank Dr. James F. Selgrade for the discussions which helped in analysis of results and improvement of the manuscript.

## Author Contributions

**Conceptualization:** Aurelio A. de los Reyes V, Johnny T. Ottesen.

**Data curation:** Brenda Lyn A. Gavina, Aurelio A. de los Reyes V, Johnny T. Ottesen.

**Formal analysis:** Brenda Lyn A. Gavina, Aurelio A. de los Reyes V, Mette S. Olufsen, Suzanne Lenhart, Johnny T. Ottesen.

**Funding acquisition:** Aurelio A. de los Reyes V, Suzanne Lenhart.

**Investigation:** Brenda Lyn A. Gavina, Mette S. Olufsen, Suzanne Lenhart.

**Methodology:** Brenda Lyn A. Gavina, Aurelio A. de los Reyes V, Mette S. Olufsen, Suzanne Lenhart, Johnny T. Ottesen.

**Project administration:** Aurelio A. de los Reyes V, Johnny T. Ottesen.

**Resources:** Johnny T. Ottesen.

**Software:** Brenda Lyn A. Gavina, Mette S. Olufsen.

**Supervision:** Aurelio A. de los Reyes V, Mette S. Olufsen, Suzanne Lenhart, Johnny T. Ottesen.

**Validation:** Brenda Lyn A. Gavina, Aurelio A. de los Reyes V, Mette S. Olufsen, Suzanne Lenhart, Johnny T. Ottesen.

**Visualization:** Brenda Lyn A. Gavina, Aurelio A. de los Reyes V.

**Writing – original draft:** Brenda Lyn A. Gavina, Aurelio A. de los Reyes V, Mette S. Olufsen, Suzanne Lenhart, Johnny T. Ottesen.

**Writing – review & editing:** Brenda Lyn A. Gavina, Aurelio A. de los Reyes V, Mette S. Olufsen, Suzanne Lenhart, Johnny T. Ottesen.

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
