## [Decision Letter · Decision Letter 0]

31 May 2022

Dear Dr. de los Reyes V,

Thank you very much for submitting your manuscript "Toward an optimal contraception dosing strategy" for consideration at PLOS Computational Biology.

As with all papers reviewed by the journal, your manuscript was reviewed by members of the editorial board and by several independent reviewers. In light of the reviews (below this email), we would like to invite the resubmission of a significantly-revised version that takes into account the reviewers' comments.

We cannot make any decision about publication until we have seen the revised manuscript and your response to the reviewers' comments. Your revised manuscript is also likely to be sent to reviewers for further evaluation.

Sincerely,

Krasimira Tsaneva-Atanasova

Guest Editor

PLOS Computational Biology

Mark Alber

Deputy Editor

PLOS Computational Biology

Reviewer's Responses to Questions

**Comments to the Authors:**

Reviewer #1: Context and aims of the work

Improving the capacity of controling the reproductive function in women without exposing them to side effects remain an important challenge worth addressing it. Hormonal contraceptives were intially developed to block the normal endocrine process leading to ovulation. It was achieved by a daily administration of a « pill » composed of synthetic compounds, ethinyl- estradiol and progestin, administered either sequentially or combined. Severe cardio- vascular effects lead to a reduction in the dosage of the estrogen component leading to the current use of « mini- pill » in the seventies. At the same period, a series of unwanted pregnancies in women exposed to the sequential pill lead to realise that any deviation from the ideal administration ot this kind of contraceptive had the opposite effect, leading to ovulation, eventually ending up with pregnancy. Since then, slight improvements have been made by using different estrogen to progestin ratios and different kinds of progestins with more or less androgenic potency.

Injections of the progestin depoprovera have been used for decades for preventing pregnancies. Modern forms of progestin- alone exposure are availaible either orally or as subdermal implant or levonorgestrel- releasing IUD. A blockade of ovulation is not necessarly achieved but a global anticonceptional effect is expected through several indirect either local or distant effects. Their acceptance by women has to be balanced with their several more or less pronounced side effects.

It is clear that only incremental innovations for hormonal contraception have been proposed in the past decades with a rising concern about the long term effects of cumulated steroid hormone exposure and the possibility of decreasing it.

The idea or using both mathematical modelling of the menstrual cycle and optimal control methods to address the issue of improving current hormonal contraceptives in order to obtain anovulation is, in this regard, somehow original.

The introduction gives a fair account of past literature about mathematical modeling of the menstrual cycle. It ends with a brief summary of the achieved results based on a modification of the model developed by Margolskee fitted on published clinical data.

The method section describes the main temporal evolution of the endocrine secretions along the menstrual cycle, including inhibin, allowing to propose some mechanisms for anovulation, directly related to FSH and estradiol circulating levels. The mathematical modelling includes only the pituitary level and the ovarian level, with several assumptions about the respective clearances of the interacting hormones.

The matlab functions used both for adjusting the full model to the dataset and for solving the optimal control problem are well described.

Comments

About the mathematical modelling of the menstrual cycle

Mathematical models of the whole endocrine reproductive female cycle has been published over the past thirty years, mostly based on systems of multi-parameterized differential equations aiming to model the complex non linear relationships between hypothalamic-, pituitary- and ovarian secretions. Some of those models address the complete aspects of the reproductive function starting from the resting follicles growing to the pre- antral follicle able to react to FSH via functional specific receptors, allowing them to enter into the final stage of maturation leading after a selection process to ovulation, with a full time scale spanning several months largely exceeding the duration of a menstrual cycle. Very few of them address the issue of the intra woman variability of a menstrual cycle not only for its entire duration but also for its respective follicular phase and luteal phase, the latter being more constant than the former, as a temporary endocrine gland, making the prediction of the ovulatory period more or less reliable either for selecting the optimal period for conceiving or for avoiding it.

Hence, both in animal studies and in clinical practice there is often a need of additional clinical biomarkers or imaging clues for better accuracy. It justified several works to optimize the monitoring of FSH and Estradiol to achieve timely ovulation both in animal studies and in humans in medically assisted procreation. Some of those animal studies are explicitely refering to optimal control theory : see for instance Clément F. et al, Math Biosci. 1998 ;152(2):123-42. Optimal control of the cell dynamics in the granulosa of ovulatory follicles; Clément F, Monniaux D. Prog Biophys Mol Biol. 2013, 113: 398-408 Multiscale modelling of ovarian follicular selection.

The authors mention some interesting works based on optimal control theory in the field of prostate cancer or type-1 diabetes. It is tempting to assume that similar methods could be applied in human reproduction to optimize the timing and dosage of the exogenous hormonal stimulations in order to achieve ovulation when it is absent or defective.

Birth control appears as a rather different issue as it concerns usually healthy women who want to avoid pregnancy over a long period in a more or less changing environment. The issue is to obtain a reliable means of avoiding pregnancy with very limited side effects, taking into account that the superimposed hormonal exposure is interfering with the normal functioning of the woman’s endocrine organs, with the possibility of escapes.

About the fit of the model and the the software used for solving the optimal control problem.

The authors used to fit their multi- parametric model the nelder- mead algorithm using a least squares error criteria. The dataset actually used is not available to the reader making not totally clear what is meant by the repeatition of the data four times and what z- scores are used.

As the Nelder- Mead algorithm can be very slow, they do not mention if they experienced convergence issues. In addition no information is given about the variance of the estimated parameters.

About solving the optimal control criteria

The objectives of minimizing the exogenous exposure to both estrogen and progestin over time in order to keep progesterone below a fixed threshold are clearly stated. Some additional explanations could have been added regarding the influence of the second and third terms in the integrand as well as the choice of the P0 value. Is the delay tau fixed or optimized in the process and how it is either fixed or initialized ?

* About the signifiance of the biological findings

The reproductive function in women is a very complex multiscale dynamical system highly exposed to both endogeous and exogenous factors. It explains both the large intra – and between women variability of the menstrual cycle length over a the reproductive period.

The reviewer was very concerned by the total absence of discussion of this aspect as the only dataset used by the authors concern daily mean values of measured blood concentrations of pooled menstrual cycles in normally cycling women. The dataset was extracted from a figure from a single article (Welt et al). No information is given how hormonal profiles from different women were scaled together. Was it scaled accordind to their first day of their respective cycles or a posteriori according to the day of the putative ovulation (LH peak) ? Hence, the model is adjusted to an ideal profile which does not correspond to any existing woman. The model would have been more convincing if fitted to the individual hormonal profiles.

The goal of any contraceptive method, whatever is the method, is to insure a protection against an unwanted pregnancy over a periode of time, not restricted to a single menstrual cycle. It is classically measured using the Pearl index corresponding to the number of failures over a two year period i.e. 24000 women x cycles assuming menstrual cycle of an ideal one month duration.

The present mathematical approach does not address this very important issue.

Furthermore, several strong assumptions are made which limit the conclusions ot the authors

- Assumption 1 : page 11, last paragraph, the statement appears in contradiction with the corresponding sentence (« the clearance of the ovarian hormones is faster than the clearance of pituitary hormones ») . Indeed the clearance of steroids is in the range of hours as compared to the one of peptide hormones which is in the range of minutes.

- Assumption 2 : during a menstrual cycle, the FSH level starts rising earlier than it is assumed in the article. It starts rising indeed at the end of the previous luteal phase and the early days of menstruation, hence allowing the growth recovery of the follicles reacting to FSH.

Failure to restart the hormonal contraceptive in time , whatever is its composition, allows the normal physiological process of selection and dominance ending up with ovulation to resume. Shortening the period and the level of exogenous hormone exposure to avoid ovulation as suggested by the authors according to their simulations, mirrors the methods and apps proposed for the natural control of ovulation, the reliability of those always assuming normally cycling women in a stable state. It contradicts the principle of a safe contraceptive means open to all women from all conditions, environmental exposures and ages.

- Assumption 3 : A precise menstrual timing is known in all women.

Several studies have estimated the probability of conception according to the day of the menstrual cycle. Of course conception is not superimposable to ovulation but it means at least that an ovulation occurred right before the conception. The probability of ovulation starts rising as early as the 6th day of the cycle and remains elevated until late in the cycle corresponding to ovulation disorders.

Ferreira-Poblete A. Adv Contracept. 1997 ;13 :83-95. The probability of conception on different days of the cycle with respect to ovulation: an overview.

Royston P. Stat Med. 1991 10:221-40. Identifying the fertile phase of the human menstrual cycle.

Lynch C et al. Paediatr Perinat Epidemiol. 2006;20 Suppl 1:3-12. Estimation of the day-specific probabilities of conception: current state of the knowledge and the relevance for epidemiological research.

Stirnemann JS et al. Hum Reproduct, 2013, 28, 110- 1116. Day-specific probabilities of conception in fertile cycles resulting in spontaneous pregnancies

All these studies strongly suggest at least an absence of a precise knowledge of the menstrual timing in a large fraction of women when initiating an hormonal contraception. It explains why the time for initiating oral contraception should be very early in the menstrual cycle in order to block the ovulation. The following « pill cycles » are indeed artificial and cannot be interpreted as menstrual cycles. The exposure to exogenous hormones induces an artificial regulation of the reproductive axis with differences from one woman to another.

- Assumption 4 : A important assumption is made regarding the effect of exogenous steroids supposed to have the same effect as endogenous natural hormones.

Actually, exogenous hormones used in contraceptives do not act totally as endogenous hormones. This is clear for progestins whose effects are different regarding both their metabolisms and their effects. Natural progesterone has strong physiological effects which differ from synthetic progestin effects used in hormonal contraceptives. The effects of the later differ according to their nature and their androgenic effects, with an impact of GnRH pulsatility and their capacity to inhibit ovulation when used alone.

Natural progesterone is available as a drug and frequently prescribed in reproductive medicine either orally or vaginally but its metabolism presents a large variability which does not allow its use for ovarian blockade.

Similarly, giving progesterone receptor modulator compounds including progestins during the follicular phase of the menstrual cycle will affect the normal course of the cycle either by suppressing or simply delaying ovulation depending of the follicular status. It remains a serious matter of concern when trying to estimate the failure of emergency hormonal contraception as many of the few unwanted pregnancies observed in this situation can be attributed to delayed ovulations after exposure to high levels of steroid hormones.

Similar observations in term of metabolism can be made about ethinyl estradiol which represents the more frequent estrogen component in hormonal oral contraceptives. Hormonal contraceptives using natural estradiol have been recentlly proposed with similar limitations.

Experiments in Rhesus monkeys have clearly shown that estradiol given in the mid or late follicular phase shuts down the GnRH pulsatility hence inhibiting ovulation (see for instance Grosser et al. Neuroendocrinology. 1993 Jan;57(1):115-9 ; O'Byrne K et al, . Neuroendocrinology. 1993 Apr;57(4):588-92.), though a similar effect has been debated in humans.

* Ethical issues : none

* Reproducibility of the results 

Except from the fact that the dataset extracted from a figure published in the article by Welt et al is not directly provided, the reproducibility of the mathematical part appears good. It is not clear what is behind the « manual adjustments » carried out after optimization. No direct code is provided.

* Substantial evidence for the proposed conclusions

In its present form, the article describes an interesting mathematical exercise based on a series of strong assumptions and a rather limited dataset corresponding to daily mean hormonal values of normally cycling women. All together, it strongly limits the possibility to translate its results to real contraceptive issues.

Reviewer #2: Please see attached file

Reviewer #3: See attached

**Have the authors made all data and (if applicable) computational code underlying the findings in their manuscript fully available?**

Reviewer #1: **No: **The dataset extracted from a figure published in the article by Welt et al is not directly provided, the reproducibility of the mathematical part appears good. The matlab functions, which were used are mentioned.

Reviewer #2: Yes

Reviewer #3: **No: **

PLOS authors have the option to publish the peer review history of their article (what does this mean?). If published, this will include your full peer review and any attached files.

Reviewer #1: **Yes: **Thalabard, Jean- Christophe

Reviewer #2: **Yes: **Urmila Diwekar

Reviewer #3: No
---

## [Decision Letter · Decision Letter 1]

18 Dec 2022

Dear Dr. de los Reyes V,

Thank you very much for submitting your manuscript "Toward an optimal contraception dosing strategy" for consideration at PLOS Computational Biology.

As with all papers reviewed by the journal, your manuscript was reviewed by members of the editorial board and by several independent reviewers. In light of the reviews (below this email), we would like to invite the resubmission of a significantly-revised version that takes into account the reviewers' comments.

Specifically, Reviewer 1 has raised significant concerns about the biological findings (see below). These need to be carefully acknowledged and discussed in the revised manuscript. Please make sure you indicate clearly the changes you have made should you decide to resubmit a revised version of your paper. I would also recommend that the usage of English language and grammar are carefully checked during the revision process.

We cannot make any decision about publication until we have seen the revised manuscript and your response to the reviewers' comments. Your revised manuscript is also likely to be sent to reviewers for further evaluation.

Sincerely,

Krasimira Tsaneva-Atanasova

Guest Editor

PLOS Computational Biology

Mark Alber

Section Editor

PLOS Computational Biology

Reviewer's Responses to Questions

**Comments to the Authors:**

Reviewer #1: * About the fit of the model and the software used for solving the optimal control problem.

The authors use to fit their multi- parametric model the nelder- mead algorithm based on a weighted least squares relative error criteria combining the five temporal profiles E(t), P(t), LH(t), FSH(t), Inh(t) . The criteria is now more explicitely expressed as well as the values of the weights corresponding to z- scores of their respective maxima.

The objectives of minimizing the exposure to both exogenous estrogen and exogenous progestin over time in order to keep progesterone below a fixed threshold are clearly stated. Some additional useful explanations are now added regarding the influence of the second and third terms in the integrand of the objective function J(u), as well as the choice of the P0 value.

The main assumption here is that the two exogenous compounds are strictly identical to the endogenous hormones and that the former are simply modifying the levels of the latter. It represents a very strong assumption which remains very far from the current reality.

* Reproducibility of the result

The authors provide now the dataset as they extracted it from a figure published in the article by Welt et al. In addition the matlab code is now accessible. Hence the reproducibility of the mathematical part appears good. In addition, the authors give some clues about what they called « manual adjustments » at the time of optimization.

* About the results

The reviewer acknowledges the explanation given by the authors about the origin of the dataset, which are now directly available. However it remains a purely artificial ideal averaged endocrine profile, scaled a posteriori according to the day of ovulation and not to the individual starting times of the menstrual cycles of the women who contributed to the study by Welt et al. Repeating four times an unique averaged profile does not give any information about the within- between women’s variabilities. It adds a spurious 28 days periodicity which is clearly not natural.

- Case 1 appears as an unrealistic, yet interesting, theoretical exercice of long term administration of estradiol. Unfortunately, such an administration will most probably induce a growth of the endometrium exposing to hyperplasia and bleeding. It justifies the need from time to time to leave an open window without estrogen to allow endometrial bleeding, though the choice of the frequency of these off- windows remains completely free..

It is interesting to note that exposing female rhesus monkeys to a constant intra- venous infusion of a low dose of estradiol has been experimented. It resulted in a complete shut down of the GnRH pulse generator which lasted several days or weeks (see for instance Ordog & Kobil, PNAS, 1995, 92, 5813 - 5136; O’Byrne & Knobil, Hum Reprod, 1993, 8 (S2), 37-40). However endometrial bleeding remains very limited in this species.

Restraining the administration of estradiol to shorter periods and varying regimens might expose to different effects not totally predictable depending on the degree of maturity of the existing developing follicles.

- Case 2 

It assumes that natural progesterone can be given exogenously with a constant concentration level over the 24- hour period. When started at the end of the previous luteal phase it should theoretically induce an inhibition of FSH release by the gonadotropin cells, and hence hinder the follicular growth with all the consequences on estradiol secretion and absence of ovulation. It is clear that this regimen has to be maintained continously as any sharp drop in the progesterone level will be followed by an increase in FSH and a resumption of the late stage of folliculare growth. Progestins are clearly different in their effects on the hypothalamo- pituitary ovarian axis as compared to natural progesterone and their effects are not uniform depending on their androgenic effect and their respective metabolism. Hence a large part of the contraceptive effect of progestin- alone contraceptive, either given orally or as implant relies on their effects on the vaginal secretion and the tubal ciliary mobility inhibition. As the follicular growth and the ovulation is not constantly prevented, any dysruption in the intake exposes to unwanted pregnancies with a higher risk of ectopic pregnancy.

- Case 3

The use of a combined administration corresponds to the majority of pill proposed on the market, although not with natural compound except for some recent pill using estradiol valerate as the estrogenic component. The originality of the work is to adjust the administration profile of both estradiol and progesterone to limit the total progesterone concentration at any time. The analysis of the outcome of the model is rather complex to understand from the physiological standpoint although it complies with the behavior of the proposed deterministic model. After a few pill cycles, it remains hard to speak about a follicular phase and a luteal phase as they do not exist anymore. However a follicular growth persists up to a follicular stage where the FSH receptors are active. The suggestion of delaying the administration of estrogen and to limit it to the mid- pill cycle exposes to an abrupt decrease in the estrogen level which may stimulate a rise in FSH with the consequence of selecting a follicle ending up with an ovulation. An increase in the amount of the exogenous natural progesterone at later stage as indicated might have the opposite effect of favoring the implantation of a fertilized ovum.

* About the signifiance of the biological findings

The reviewer re- emphasizes that the reproductive function in women is a very complex multiscale dynamical system highly exposed to both endogeous and exogenous factors. It explains both the large intra – and between women variability of the menstrual cycle length over a the reproductive period. The reviewer remains very concerned by a limited discussion about this aspect as the only dataset used by the authors concern daily mean values of measured blood concentrations of pooled menstrual cycles in normally cycling women. The authors proposes a sensitivity analysis which does not fully address the intrinsic random aspects of the endocrine cycle, as the only relative « constant » part of the cycle appears to be the duration of the luteal phase when an ovulation occurred, whereas the process of selection of the dominant follicles and its subsequent growth induces a more random follicular phase duration.

The claim that this modelling represents a proof of concept that could be adapted to individual profiles is in theory correct but obtaining this daily blood profile even for one single cycle in routine appears totally unrealistic for both financial and practical reasons. Suggesting that the administration regimen could be triggered by estradiol blood concentration appears as a still futuristic and unrealistic perspective. It appears at least very remote of what is the current concern of access to contraception for all the women whatever their socio- economic status is.

The goal of any contraceptive method, whatever is the method, is to insure a protection against an unwanted pregnancy over a periode of time, not restricted to a single menstrual cycle with women exposed to various environments, including stresses, infections, acute metabolic conditions, ....

The present manuscript does not really address this very important issue.

Furthermore, several strong assumptions are made which limit the conclusions ot the authors. The explanations given in their rebuttal to the three assumptions, i.e. both within- and between- women’s variabilities, modification in the dosage and timing along time, failure of taking into account the exact nature of the administered hormones and their metabolisms either alone or combined are not enough cautious about their consequences on the interpretation of their results.

Hence any suggestion of shortening the duration of exposure to exogenous steroids exposes to the risk of unwanted pregnancies. This has been unfortunately largely reported with the consequence of always shortening the pill- off period and not extending it. The exact nature of the steroid compounds is also very important both in terms of actions on the steroid receptors and in terms of metabolism. Natural hormones cannot be so easily used so far for contraception. Any change of dosage along the cycle exposes to the risk of confusion in the daily intake, unless depending on a automatized rather sophisticated delivery system which again appears so far nrealistic.

* Minor comments

page 17/23, the paragraph « Hormonal contraception benefits go beyond contraception….Estrogen also decrease motor skills, reducing the normal neurosmuscular protective mechanisms of the knee » does not appear very relevant to the present work and could be easily suppressed. Intensive training more or less associated with severe nutritional restriction expose indeed to hypo-estrogenemia and its pathological consequences. Indeed, an adapted hormonal contraception bringing back a normo- estrogenic state alleviates these effects., without the need of a sophisticated modelling approach.

* Substantial evidence for its conclusions

In its present form, the article describes an interesting mathematical exercise applying some results of control theory to a hyper-parametrized deterministic model. They deserve credit for this aspect. It is however based on both a series of strong assumptions more or less realistic and a very limited ideal « real » dataset which limits the possibility to translate these results to real contraceptive issues. Therefore, their conclusions in their formulation do not appear cautious enough to be shared to a wide audience who might be confused and impressed by the mathematical developments..

Reviewer #2: All comments are addressed.

**Have the authors made all data and (if applicable) computational code underlying the findings in their manuscript fully available?**

Reviewer #1: Yes

Reviewer #2: Yes

PLOS authors have the option to publish the peer review history of their article (what does this mean?). If published, this will include your full peer review and any attached files.

Reviewer #1: **Yes: **JC Thalabard

Reviewer #2: **Yes: **Urmila Diwekar
---

## [Editor Report · Decision Letter 2]

27 Feb 2023

Dear Dr. de los Reyes V,

We are pleased to inform you that your manuscript 'Toward an optimal contraception dosing strategy' has been provisionally accepted for publication in PLOS Computational Biology.

Best regards,

Krasimira Tsaneva-Atanasova

Guest Editor

PLOS Computational Biology

Mark Alber

Section Editor

PLOS Computational Biology

---

## [Editor Report · Acceptance letter]

23 Mar 2023

PCOMPBIOL-D-22-00522R2 

Toward an optimal contraception dosing strategy

Dear Dr de los Reyes V,

I am pleased to inform you that your manuscript has been formally accepted for publication in PLOS Computational Biology. Your manuscript is now with our production department and you will be notified of the publication date in due course.

With kind regards,

Anita Estes
